# Activation of melanocortin-1 receptor signaling in melanoma cells impairs T cell infiltration to dampen antitumor immunity

Yazhong Cui[1,2,8], Yang Miao[2,3,8], Longzhi Cao[1,2], Lifang Guo[4], Yue Cui[2,5], Chuanzhe Yan[2,6], Zhi Zeng [1,2], Mo Xu [1,2,7,9] ✉ & Ting Han [1,2,7,9] ✉

Inhibition of T cell infiltration dampens antitumor immunity and causes resistance to immune checkpoint blockade (ICB) therapy. By in vivo CRISPR screening in B16F10 melanoma in female mice, here we report that loss of melanocortin-1 receptor (MC1R) in melanoma cells activates antitumor T cell response and overcomes resistance to ICB. Depletion of MC1R from another melanocytic melanoma model HCmel1274 also enhances ICB efficacy. By activating the GNAS-PKA axis, MC1R inhibits interferon-gamma induced *CXCL9/10/11* transcription, thus impairing T cell infiltration into the tumor microenvironment. In human melanomas, high *MC1R* expression correlates with reduced *CXCL9/10/11* expression, impaired T cell infiltration, and poor patient prognosis. Whereas MC1R activation is restricted to melanoma, GNAS activation by hotspot mutations is observed across diverse cancer types and is associated with reduced *CXCL9/10/11* expression. Our study implicates MC1R as a melanoma immunotherapy target and suggests GNAS-PKA signaling as a pan-cancer oncogenic pathway inhibiting antitumor T cell response.

The immune system interfaces closely with tumors during the entire process of disease progression, ranging from early tumor formation to metastasis[1]. Effective antitumor immunity relies on both the innate and adaptive immune systems, and involves multiple steps including the processing and presentation of tumor antigens by antigen-presenting cells (APCs), activation of antigen-specific T cells, trafficking of effector T cells into tumors followed by recognition and elimination of cancer cells by T cells[2]. With disease progression, tumors can develop multiple mechanisms to evade immunosurveillance. Such mechanisms include the selection of tumor variants avoiding immune recognition (referred to as immunoediting), the production of immune suppressive mediators within the tumor microenvironment, the resistance to killing by tumor reactive cytotoxic T lymphocytes (CTLs), as well as the inhibition of immune cell accumulation in tumors[2–4].

The understanding of tumor immune evasion mechanisms has resulted in immunotherapeutic agents that yielded remarkable clinical efficacy in cancer treatment. For example, immune checkpoint blockade (ICB) therapy targeting the interaction between the programmed cell death protein 1 (PD-1)/PD-L1 receptor-ligand pair has proven to be highly effective in many cancer types, such as melanoma, non-small cell lung cancer, head and neck squamous cell cancer, and solid tumors deficient in mismatch repair[5]. However, the therapeutic benefit of ICB is limited to a subset of patients in each cancer type despite the use of predictive biomarkers such as PD-L1 expression,

[1]Graduate School of Peking Union Medical College and Chinese Academy of Medical Sciences, 100730 Beijing, China. [2]National Institute of Biological Sciences, 102206 Beijing, China. [3]PTN Joint Graduate Program, School of Life Sciences, Tsinghua University, 100084 Beijing, China. [4]Department of Thoracic Surgery, Beijing Chaoyang Hospital, Capital Medical University, 100020 Beijing, China. [5]Graduate Program, School of Life Sciences, Beijing Normal University, 100875 Beijing, China. [6]PTN Joint Graduate Program, School of Life Sciences, Peking University, 100871 Beijing, China. [7]Tsinghua Institute of Multidisciplinary Biomedical Research, Tsinghua University, 102206 Beijing, China. [8]These authors contributed equally: Yazhong Cui, Yang Miao. [9]These authors jointly supervised this work: Mo Xu, Ting Han. ✉e-mail: xumo@nibs.ac.cn; hanting@nibs.ac.cn

tumor mutation burden, and mismatch repair deficiency[6–8]. Thus, expanding our knowledge on additional mechanisms that promote tumor immune evasion remains crucial for the improvement of cancer immunotherapies.

Recent basic and clinical studies have revealed a plethora of factors that influence antitumor immunity, among which the level of T cell infiltration has been shown to be highly correlated with response to ICB treatment[9]. T cell infiltration is an essential step of the antitumor immune response[4]. To find tumor cells to attack, T cells must extravasate through blood vessels, traverse through the extracellular matrix, and infiltrate into the tumor parenchyma[10]. Previous studies have revealed several molecular mechanisms employed by cancer cells to impair T cell infiltration, such as activation of WNT/β-catenin signaling to impair dendritic cell recruitment, epigenetic silencing of chemokine expression to block T cell migration, and upregulation of TGF-β signaling to trap T cells in the tumor stroma[11–13]. As cancer is a collection of diseases driven by diverse genetic alterations[14] and evolved from distinctive tissue microenvironments[15], it is highly likely that additional drivers inhibiting T cell infiltration remain to be discovered.

G protein-coupled receptors (GPCR) belong to a large family of cell-surface receptors that regulate essential physiological functions, such as environmental sensing, homeostatic regulation, and immune defense[16]. GPCR activation involves agonist binding at the extracellular side, causing a conformational change in the receptor[17]. Agonist-bound GPCRs then couple to the heterotrimeric G proteins, promoting the exchange of GDP for GTP at the Gα subunit[18]. The signaling properties of GPCRs are defined by coupling to one of four class of Gα proteins (Gαs, Gαi/o, Gαq/11, and Gα12/13), which bind to and stimulate their cognate downstream effectors to mediate GPCR signaling[19]. Cancer cells can hijack GPCR signaling to promote tumorigenesis[20–22]. Overexpression of diverse GPCRs in cancer tissues are correlated with poor patient prognosis[23,24]. In addition, mutations that constitutively activate Gαs and Gαq/11 have been observed in cancers arising from the pituitary gland, colon, pancreas, kidney, liver, and eyes[20,25–27]. Whereas previous studies have revealed the mechanisms by which aberrant GPCR signaling promote cancer cell proliferation, survival, invasion, and metastasis, their roles in modulating antitumor immunity are gaining more attention[28–30].

In this work, we perform pooled in vivo CRISPR knockout screening using the mouse B16F10 melanoma model and discover that depletion of melanocortin-1 receptor (MC1R) in melanoma cells activates antitumor T cell response. Genetic analysis reveals that MC1R signals through the GNAS-PKA axis to repress the transcription of chemokine genes *CXCL9/10/11*, resulting in impaired T cell infiltration into the tumor microenvironment. We further validate these findings in another melanoma model HCmel1274. Extending from findings in melanoma, we demonstrate that oncogenic mutations that constitutively activate GNAS repress *CXCL9/10/11* expression to dampen antitumor T cell response across human cancer types, including ICB-refractory pancreatic and liver cancers. These results implicate GPCR-GNAS-PKA signaling as a pan-cancer oncogenic pathway impairing T cell infiltration.

## Results

### In vivo CRISPR screening identifies *Mc1r* as a candidate immune evasion gene of B16F10 melanoma

To model tumors lacking T cell infiltration in the tumor microenvironment (TME), we used the mouse melanoma cell line B16F10, which forms tumors with low immune cell infiltration and evades immunosurveillance when transplanted into syngeneic mice[31]. Seeking to reveal the genetic drivers of immune evasion in B16F10, we performed pooled CRISPR knockout screening to identify genes that were essential for the growth and/or survival of transplanted B16F10 cells in vivo in the presence of a complete immune system but dispensable for the growth and/or survival of cultured cells in vitro (Fig. 1A).

Compared to a previous study that employed in vivo CRISPR screening to identify highly expressed genes that help B16F10 evade immunosurveillance[32], our screening examined 6,053 genes encoding membrane-associated or secreted proteins[33] as these were more likely to mediate the communication between cancer cells and immune cells in the TME (Supplementary Fig. S1A). We transduced our custom sgRNA libraries into B16F10 melanoma cells stably expressing Cas9, and then cultured the transduced cells in vitro or transplanted them into the dorsal flanks of wild-type mice for tumor growth in vivo (Fig. 1A). After two weeks, we harvested 6–8 tumors from sacrificed mice and used Illumina sequencing to compare their sgRNA representations to those of B16F10 cells cultured in vitro. By applying the MAGeCK (Model-based Analysis of Genome-wide CRISPR/Cas9 Knockout) algorithm[34] to rank genes according to their sgRNA depletion in vivo relative to in vitro, we identified genes encoding known mediators of immune evasion (*Cd47* and *Ptpn2*)[32,35]. We also identified known suppressors of tumor growth (*Nf2* and *Kirrel*)[36]. In addition, we identified *Mc1r* as a candidate gene mediating immune evasion in B16F10 melanoma (Fig. 1B, C, Supplementary Fig. S1B, and Supplementary Data 1).

### Depletion of MC1R enhances antitumor immunity against B16F10 melanoma

The *Mc1r* gene encodes the G protein-coupled receptor (GPCR) melanocortin-1 receptor, which is primarily located on the surface of melanocytes and plays an essential role in skin pigmentation in mammals[37]. Upon binding to its ligands, such as α-melanocyte-stimulating hormone (α-MSH), MC1R activates the adenylyl cyclase via G protein Gαs (GNAS), thus triggering cyclic adenosine 3′,5′-monophosphate (cAMP) production and protein kinase A (PKA) activation, leading to the phosphorylation and activation of a downstream transcription factor cAMP response element-binding protein (CREB)[38]. Based on RNA sequencing (RNA-seq), *Mc1r* was the only highly expressed Gαs-coupled GPCR in B16F10 cells (Supplementary Fig. S1C).

To deplete MC1R from B16F10 cells, we used a nuclease-dead Cas9 (dCas9) variant fused to a KRAB transcription repressor[39] to inhibit the transcription of *Mc1r*. By qPCR, we observed greater than 100-fold reduction of *Mc1r* gene expression in B16F10-dCas9 cells expressing an *Mc1r* sgRNA compared to those expressing a non-targeting control (NTC) sgRNA (Fig. 1D). To ensure the inhibition of *Mc1r* transcription could result in defective MC1R function, we monitored CREB phosphorylation by western blotting and CREB activation by the cAMP responsive element-firefly luciferase (CRE-Fluc) reporter assay[40]. In B16F10-dCas9 cells expressing the NTC sgRNA, activation of MC1R by α-MSH stimulated the phosphorylation of two CREB family proteins, CREB1 and activating transcription factor 1 (ATF1), and increased CRE-Fluc reporter activity (Fig. 1E, F). Expression of the *Mc1r* sgRNA blocked α-MSH-induced CREB1/ATF1 phosphorylation in B16F10-dCas9 cells (Fig. 1E). Moreover, B16F10-dCas9 cells expressing the *Mc1r* sgRNA displayed a lower basal CRE-Fluc reporter activity, which was no longer responsive to α-MSH stimulation (Fig. 1F). These results demonstrate that we have sufficiently repressed *Mc1r* expression to impair MC1R function in B16F10 cells.

In order to examine whether MC1R promotes immune evasion of B16F10 melanoma, we transplanted B16F10-dCas9 cells transduced with NTC or *Mc1r* sgRNA into mice and monitored tumor growth and host survival over time. In wild-type mice, MC1R depletion slowed B16F10-dCas9 tumor growth, leading to a significant survival advantage for tumor-bearing mice. In contrast, MC1R depletion did not affect tumor growth or host survival in immunodeficient *NOD-Prkdc-Il2r* gamma (NCG) mice or T cell receptor beta (TCRβ) knockout mice (Fig. 1G). As B16F10 melanoma is known to be resistant to anti-PD-1 treatment[31], we examined whether MC1R depletion could overcome resistance to anti-PD-1. Whereas anti-PD-1 treatment resulted in a low

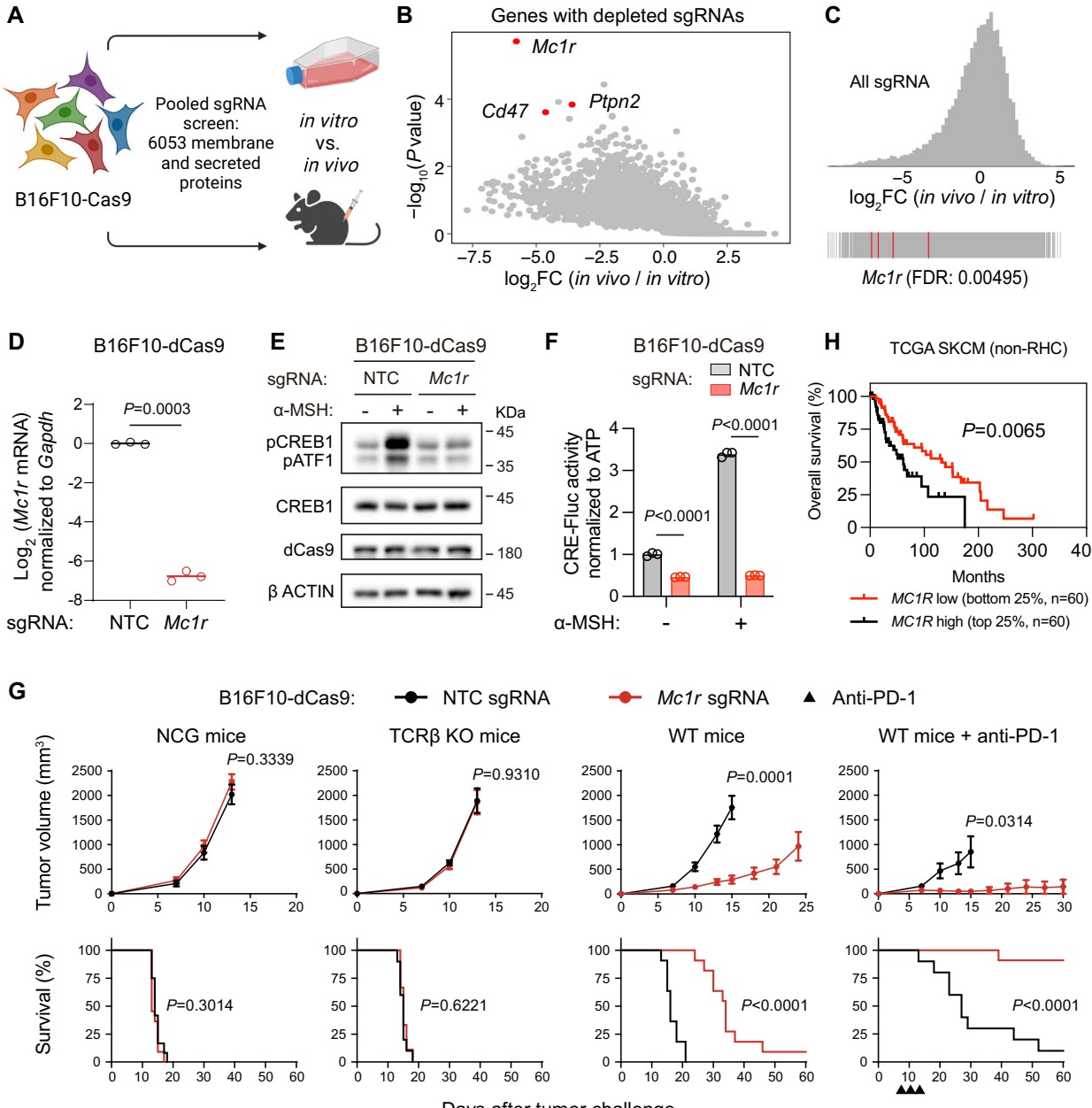

**Fig. 1 | In vivo CRISPR screening identifies *Mc1r* as a mediator of immune evasion in B16F10 melanoma. A** Strategy for in vivo CRISPR screening. Created with biorender.com. **B** Scatterplot depicting genes with significantly depleted sgRNAs in B16F10-Cas9 tumors ($n = 8$, 6, and 7 independent tumors respectively for sublibraries A, B, and C) versus cultured B16F10-Cas9 cells. $P$ values were computed by MAGeCK from the negative binomial model using a modified robust ranking aggregation algorithm. **C** Frequency histogram of $\log_2$FC of all sgRNAs in B16F10-Cas9 tumors ($n = 8$, 6, and 7 independent tumors respectively for sub-libraries A, B, and C) compared to cultured B16F10-Cas9 cells. *Mc1r*-targeting sgRNAs are shown by the red lines. **D** qPCR quantification of *Mc1r* mRNA levels in B16F10-dCas9 cells transduced with non-targeting control (NTC) sgRNA or *Mc1r* sgRNA. Lines indicate the mean of three independent samples. The Welch's $t$ test (two-tailed) was used to determine statistical significance. **E** Western blotting of the indicated proteins in B16F10-dCas9 cells transduced with NTC sgRNA or *Mc1r* sgRNA. Cells were untreated or treated with 1 μM α-MSH for 1 h. A representative result was shown from two independent experiments. Uncropped western blot images are provided as a Source Data file. **F** CRE-Fluc reporter activity of B16F10-dCas9 cells transduced

with NTC sgRNA or *Mc1r* sgRNA. Cells were untreated or treated with 1 μM α-MSH for 24 h. Lines indicate the mean of three independent samples. Student's $t$ tests (two-tailed) were used to determine statistical significance. **G** Tumor volume and survival analysis of mice with indicated genotypes transplanted with B16F10-dCas9 tumors expressing NTC sgRNA or *Mc1r* sgRNA. Anti-PD-1 treatments are indicated as black triangles. Data are the mean ± s.e.m. (NCG mice: $n = 12$ for NTC sgRNA, 11 for *Mc1r* sgRNA; TCRβ KO mice: $n = 10$ for NTC sgRNA, 9 for *Mc1r* sgRNA; WT mice: $n = 11$ for NTC sgRNA, 11 for *Mc1r* sgRNA; WT mice+anti-PD-1: $n = 10$ for NTC sgRNA, 11 for *Mc1r* sgRNA). Welch's $t$ tests (two-tailed) and log-rank (Mantel−Cox) tests were respectively used to determine the statistical significance of the differences in tumor volume and survival. **H** Kaplan–Meier survival analyses among TCGA skin cutaneous melanoma (SKCM) patients (with the ethnicity of 'white', and excluding individuals with two strong red-hair-color (RHC) alleles) stratified by quartiles of *MC1R* expression level. Comparisons were made between the highest quartile versus the lowest quartile. The log-rank (Mantel−Cox) test was used to determine the statistical significance of the difference in patient survival. Source data are provided as a Source Data file.

rejection rate of control tumors (1/10), the majority of MC1R-depleted tumors (9/10) were rejected following anti-PD-1 treatment (Fig. 1G). To corroborate these findings, we used the Cas9 system to delete *Mc1r* from B16F10 cells. In comparison to two control sgRNAs targeting nonessential genes *Rosa26* and *H11*, two independent sgRNAs targeting *Mc1r* blocked α-MSH-induced CREB1/ATF1 phosphorylation and slowed tumor growth (Supplementary Fig. S1D, E).

Because recent studies have revealed the immunogenicity of Cas9 that could influence the antitumor immune response[41], we monitored dCas9 expression levels by western blotting and found that MC1R depletion did not affect dCas9 expression (Fig. 1E). Moreover, we transplanted B16F10-dCas9 cells into the Cas9 transgenic mice and observed that *Mc1r* depletion slowed B16F10-dCas9 tumor growth (Supplementary Fig. S1F). Moreover, both the in vitro and in vivo effects of MC1R depletion could be rescued by re-expression of the *MC1R* cDNA (Supplementary Fig. S1G, H). Taken together, these observations indicate that depletion of MC1R from B16F10 cells enhances the antitumor immune responses against B16F10 melanoma.

In the human population, *MC1R* red-hair-color (RHC) variants significantly impairs MC1R function[42]. A previous analysis of skin cutaneous melanoma (SKCM) patients with stated ethnicity of 'white' from the Cancer Genome Atlas (TCGA) revealed that 8% of individuals had two strong RHC alleles[43]. We used the RHC annotations from the study, excluded the individuals with biallelic RHC variants, and grouped the remaining individuals (non-RHC) according to *MC1R* expression. We found that patients with high *MC1R* expression in their tumors exhibited significantly shorter overall survival than patients with low *MC1R* expression (Fig. 1H). This result suggests that MC1R-mediated immune evasion may be conserved in human melanoma.

### Depletion of MC1R activates antitumor T cell response in B16F10 melanoma

To identify the cellular mechanisms by which MC1R drives immune evasion, we used flow cytometry to quantify infiltrated immune cell subsets in B16F10-dCas9 tumors expressing *Mc1r* sgRNA versus NTC sgRNA (Supplementary Fig. S2A). Immune cell (indicated as CD45⁺) infiltration was markedly higher in B16F10-dCas9 tumors expressing *Mc1r* sgRNA relative to NTC sgRNA, with significantly increased numbers of CD8⁺ and CD4⁺ T cells as well as natural killer (NK) cells (Fig. 2A). Among CD45⁺ cells, the percentages of CD8⁺ T cells were increased in B16F10-dCas9 tumors expressing the *Mc1r* sgRNA relative to the NTC sgRNA (Supplementary Fig. S2B). Consistently, immunofluorescence staining of CD8 revealed a significant increase of CD8-positive cells in tumors expressing *Mc1r* sgRNA, which diffusely infiltrated into the tumor parenchyma (Fig. 2B, C). These results indicate that depletion of MC1R increases T cell infiltration into the TME of B16F10 melanoma.

As CD8⁺ T cells play an essential role in antitumor immunity[44], we examined the effector functions of these cells in B16F10-dCas9 tumors by staining for tumor necrosis factor alpha (TNFα), interferon-gamma (IFNγ), and granzyme B (GZMB) following restimulation. Whereas the percentages of TNFα-, IFNγ-, or GZMB-positive cells among CD8⁺ T cells were not significantly different between the two groups, the expression levels—indicated by mean fluorescence intensities (MFI)—of IFNγ and GZMB were modestly increased in CD8⁺ T cells from tumors expressing *Mc1r* sgRNA relative to NTC sgRNA (Fig. 2D). As myeloid cells could play both anti- and pro-tumor roles and modulate the activities of T cells, we further analyzed the representative cell types in the myeloid compartment[32] in B16F10-dCas9 tumors expressing the *Mc1r* sgRNA versus the NTC sgRNA (Supplementary Fig. S3A). We observed no statistically significant difference between the two groups (Supplementary Fig. S3B). Taken together, we conclude that depletion of MC1R activates antitumor T cell response in B16F10 melanoma.

To evaluate the impact of *MC1R* on T-cell infiltration and activity in human melanoma, we examined the expression levels of multiple lineage (*CD45*, *CD3D/E/G*, and *CD8A/B*) and phenotypic markers (*PRF1*, *GZMA*, *GZMH*, *GZMK*, *KLRK1*, *NKG7*, *IFNG*, *CD69*, *CD96*, and *LAG3*) of CD8⁺ cells in *MC1R*-high versus *MC1R*-low non-RHC melanomas. We observed that *MC1R*-high melanomas were more likely to be low in T-cell infiltration and effector functions in comparison to *MC1R*-low melanomas (Fig. 2E). This result suggests that MC1R may also negatively regulate antitumor T cell response in human melanoma.

### MC1R activation inhibits the transcription of a subset of IFNγ-induced genes by repressing their promoter and enhancer elements

To investigate the molecular mechanisms by which MC1R promotes melanoma immune evasion, we performed RNA-seq to compare the transcriptomes of B16F10-dCas9 cells transduced with the *Mc1r* sgRNA versus the NTC sgRNA in the absence or presence of α-MSH. Because IFNγ is an inflammatory cytokine that broadly exists in the TME and orchestrates antitumor responses[45], we also compared the transcriptomes of the two groups following simultaneous treatment with IFNγ and α-MSH (Fig. 3A). Gene ontology analysis of differentially expressed genes in the untreated or α-MSH–treated groups did not reveal terms associated with immune-related functions (Supplementary Fig. S4A). Similarly, when treated with IFNγ and α-MSH, genes downregulated in MC1R-depleted cells relative to control cells were also not immune-related (Supplementary Fig. S4B). In contrast, when treated with IFNγ and α-MSH, MC1R-depleted cells expressed markedly higher levels of chemokine genes (*Cxcl9*, *Cxcl10*, and *Cxcl11*) and MHC class II genes (*CD74*, *H2-Aa*, *H2-Ab1*, and *H2-Eb1*) than control cells (Fig. 3A, B). Gene ontology analysis indicates that these upregulated genes in MC1R-depleted cells relative to control cells are involved in response to IFNγ, antigen presentation via MHC class II, and immune cell chemotaxis (Supplementary Fig. S4C). In contrast, neither basal nor IFNγ-induced expression of PD-L1 were affected by MC1R depletion (Supplementary Fig. S5A–C).

To identify the cis-regulatory elements of IFNγ-induced genes repressed by MC1R activation, we used Assay for Transposase-Accessible Chromatin using sequencing (ATAC-seq) to examine alterations in chromatin opening in B16F10 cells following different combinations of IFNγ and α-MSH treatment (Supplementary Fig. S6A). *CXCL9*, *CXCL10*, and *CXCL11* reside at the same genomic locus. Out of three regions in this locus that gained accessibility after IFNγ treatment, only one was repressed by α-MSH treatment. This region encompassed the last exon of *CXCL10* and ChIP-seq revealed reduced histone H3 lysine27 acetylation (H3K27ac) of this region, suggesting its role as an enhancer element underlying MC1R-mediated repression of *CXCL9*, *CXCL10*, and *CXCL11* transcription (Fig. 3C). Inspection of ATAC-seq and H3K27ac ChIP-seq peaks also identified promoter and enhancer elements that underlie MC1R-mediated repression of *CD74*, *H2-Aa*, *H2-Ab1*, and *H2-Eb1* (Supplementary Fig. S6B–D). To identify potential transcription factors that binds to the cis-regulatory elements repressed by MC1R activation, we identified enriched motif in regions that became less accessible in B16F10 cells treated with IFNγ and α-MSH in comparison to cells treated with IFNγ. The top enriched motifs were predominantly E26 transformation-specific (ETS) binding motifs, suggesting that MC1R activation interferes with ETS family of transcription factors at these regions (Supplementary Fig. S6E).

### MC1R mediates immune evasion of B16F10 melanoma by repressing *Cxcl9/10/11* expression

CXCL9, CXCL10, and CXCL11 are chemokines mediating the recruitment of CXCR3-expressing immune cells, such as type 1 T helper (Th1) cells, CD8⁺ T cells, and NK cells, into the TME[46]. By qPCR, we observed that IFNγ treatment resulted in robust induction of *Cxcl9/10/11* expression in B16F10-dCas9 cells transduced with the NTC sgRNA, which was significantly repressed when α-MSH was added simultaneously with IFNγ. In contrast, B16F10-dCas9 cells transduced with the

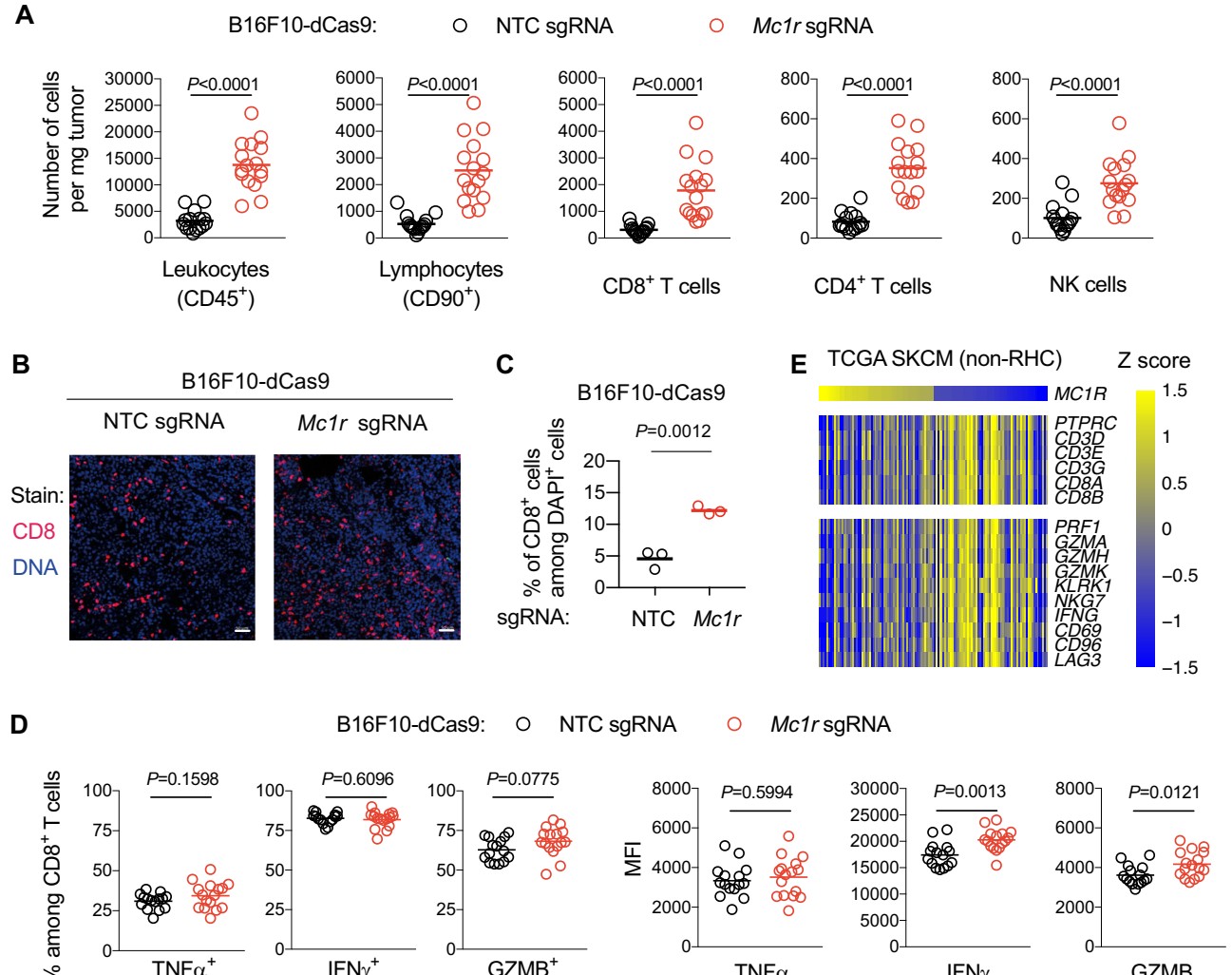

**Fig. 2 | Depletion of MC1R activates T cell response in B16F10 melanoma.**
**A** Number of tumor-infiltrating immune cell subsets per mg tumor analyzed by flow cytometry comparing B16F10-dCas9 tumors expressing NTC sgRNA ($n = 15$) versus *Mc1r* sgRNA ($n = 16$). Lines indicate the mean. Welch's *t* tests (two-tailed) were used to determine statistical significance. **B** Immunofluorescence staining of CD8-positive cells in B16F10-dCas9 tumors expressing NTC sgRNA or *Mc1r* sgRNA. DAPI staining was used to visualize nuclei. Scale bar: 50 μm. **C** Quantification of data in (**B**). Each data point is the average of CD8-positive cell densities from nine randomly chosen imaging fields of a tumor. Lines indicate the mean of three tumors. The Student's *t* test (two-tailed) was used for statistical analysis. **D** Intracellular staining of TNFα, IFNγ, and Granzyme B (GZMB) of CD8+ T cells from B16F10-dCas9 tumors expressing NTC sgRNA ($n = 15$) or *Mc1r* sgRNA ($n = 16$). Cells were re-stimulated with phorbol 12-myristate 13-acetate (PMA) and ionomycin. Frequency of cells expressing indicated molecules and their mean fluorescence intensities (MFI) were compared between the two groups. Lines indicate the mean. Welch's *t* tests (two-tailed) were used to determine statistical significance. **E** Heatmap showing Z-score normalized expression levels of indicated T cell marker genes in *MC1R*-high ($n = 62$) versus *MC1R*-low ($n = 62$) non-RHC melanomas in TCGA stratified by *MC1R* expression. Comparisons were made between the highest quartile versus the lowest quartile. Source data are provided as a Source Data file.

*Mc1r* sgRNA expressed higher levels of *Cxcl9/10/11* upon IFNγ treatment, which could not be suppressed by α-MSH treatment (Fig. 4A). By the enzyme-linked immunosorbent assay (ELISA), we observed changes in secreted CXCL9 and CXCL10 proteins from B16F10-dCas9 cells that were consistent with changes in their mRNA levels (Fig. 4B).

To test whether MC1R is active in tumors, we transplanted B16F10 cells harboring the CRE-Fluc reporter and observed reduced *Fluc* expression in MC1R-depleted tumors (Supplementary Fig. S7A). To examine whether MC1R represses *Cxcl9/10/11* expression in transplanted B16F10-dCas9 tumors in vivo, we performed RNA-seq of isolated cancer cells from B16F10-dCas9 tumors expressing the *Mc1r* sgRNA versus the NTC sgRNA. We marked B16F10-dCas9 cells with the zsGreen fluorescent protein and engrafted them into wild-type mice. After 12 days, zsGreen-positive cancer cells were sorted from dissociated tumors by fluorescence-activated cell sorting (FACS) followed by RNA-seq analysis (Fig. 4C). Similar to findings in vitro, significantly higher levels of *Cxcl9/10/11* were observed in cancer cells isolated from

MC1R-depleted tumors relative to control tumors (Fig. 4D). Moreover, the upregulation of *Cxcl9/10/11* in MC1R-depleted cancer cells resulted in significantly higher levels of *Cxcl9/10/11* in the bulk MC1R-depleted tumors (Supplementary Fig. S7A). Taken together, we conclude that MC1R represses the expression of *Cxcl9*, *Cxcl10*, and *Cxcl11* in B16F10 melanoma.

Numerous cellular sources of CXCL9/10/11 have been described in the TME, including cancer cells, tumor-infiltrating antigen-presenting cells (APCs), and inflammatory fibroblasts[46]. We therefore tested whether inhibition of cancer cell-derived *Cxcl9/10* expression by MC1R signaling is sufficient to mediate immune evasion of B16F10 melanoma. *Cxcl11* was omitted because of a 2-nucleotide insertion in its first exon in the C57BL/6 background, which results in a premature stop codon[47]. Using CRISPR/Cas9, we obtained two independent *Cxcl9/10* knockout clones of B16F10-Cas9 (Supplementary Fig. S7B). We transduced these two *Cxcl9/10* knockout B16F10-Cas9 clones with the NTC sgRNA or the *Mc1r* sgRNA and transplanted the resulting cells into wild-

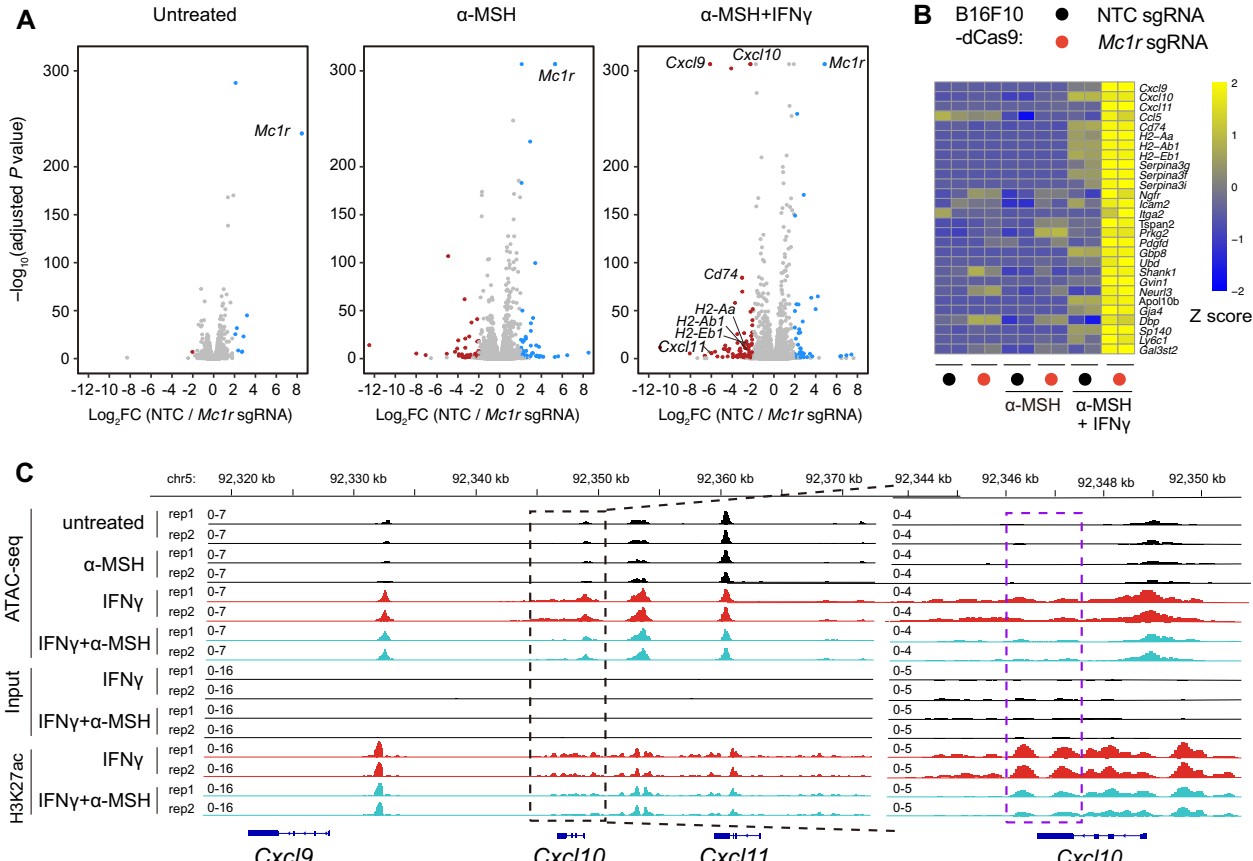

**Fig. 3 | MC1R activation represses the transcription of a subset of IFNγ-induced genes. A** RNA-seq volcano plots depicting differentially expressed genes (computed by DESeq2) in B16F10-dCas9 cells expressing NTC sgRNA versus *Mc1r* sRNA with indicated treatment of α-MSH (1 μM) and IFNγ (10 ng/ml) for 12 h. Genes differentially expressed (adjusted *P* value < 0.05) were labeled as red (log$_2$FC <− 2) or blue (log$_2$FC > 2) dots. Two independent samples were included for analysis. *P* values were calculated by Wald test using Benjamini and Hochberg correction.

**B** Heatmap (log$_2$(TPM + 1), Z-score normalized, TPM stands for transcripts per million) showing IFNγ-induced genes that were upregulated in B16F10-dCas9 cells expressing *Mc1r* sgRNA relative to NTC sgRNA. Source data are provided as a Source Data file. **C** ATAC-seq and H3K27ac ChIP-seq tracks showing changes in chromatin accessibility and H3K27ac at the *Cxcl9/10/11* locus in B16F10 cells with indicated treatment of α-MSH (1 μM) and IFNγ (10 ng/ml) for 12 h. Two independent samples were included for analysis.

type mice. Deleting *Mc1r* (by expressing the *Mc1r* sgRNA) from these two clones did not result in statistically significant difference in tumor growth or animal survival relative to control (Fig. 4E and Supplementary Fig. S7C). In contrast, deleting *Mc1r* from parental B16F10-Cas9 significantly slowed tumor growth (Fig. 4E). Western blotting showed that *Mc1r* depletion blocked α-MSH-induced CREB1/ATF1 phosphorylation in both parental and *Cxcl9/10* knockout B16F10-Cas9 cells, consistent with the role of CXCL9 and CXCL10 as downstream effectors of MC1R activation (Supplementary Fig. S7D). Taken together, these results demonstrates that the repression of *Cxcl9/10* expression by MC1R is critical for B16F10 melanoma to evade immunosurveillance.

### *MC1R* expression negatively correlates with *CXCL9/10/11* expression in human melanoma

To explore whether MC1R also represses *CXCL9/10/11* transcription in human melanoma, we compared the transcriptomes of *MC1R*-high versus *MC1R*-low non-RHC melanoma cases in TCGA. Known melanocytic genes *PMEL*, *TYR*, *MLANA*, and *TYRP1*[38] were expressed at higher levels in the *MC1R*-high group compared with the *MC1R*-low group, which is consistent with the function of MC1R in promoting melanogenesis (Fig. 4F). This result also indicates that *MC1R*'s expression levels correlate with its signaling activities. We next focused on the chemokine genes *CXCL9/10/11*. Their expression levels were indeed significantly lower in the *MC1R*-high group compared with the *MC1R*-low group (Fig. 4F and Supplementary Fig. S7E). It has been reported

that MC1R-mediated melanogenesis protects skin cells from UV damage[48]. We thus examined whether *MC1R* expression correlates with tumor mutation burden and found no statistically significant correlation (Supplementary Fig. S7F). Taken together, high *MC1R* expression correlates with low *CXCL9/10/11* expression independent of tumor mutation burden in human melanoma.

### MC1R promotes immune evasion of the mouse melanocytic melanoma HCmel1274

We explored the relevance of MC1R in human cancers by examining the expression profiles of *MC1R* in 33 human cancer types versus normal tissues using TCGA and the Genotype-Tissue Expression (GTEX) RNA-seq datasets. This analysis revealed a subset of skin cutaneous melanomas (SKCM) with *MC1R* expression higher than any other cancer types or normal tissues (Fig. 5A). This result suggests that MC1R could be an immunotherapy target for a subset of melanoma patients with high *MC1R* expression.

A panel of four syngeneic melanoma models has been reported recently, which represented a variety of molecular and phenotypic subtypes of human melanomas and exhibited diverse range of responses to immune checkpoint blockade[49]. Based on published RNA-seq data, one of the four models (M3/HCmel1274, representing the melanocytic subtype of melanomas) expressed high levels of *MC1R*. We used the dCas9 system to efficiently deplete MC1R from HCmel1274 (Fig. 5B). In HCmel1274-dCas9 cells with an NTC sgRNA,

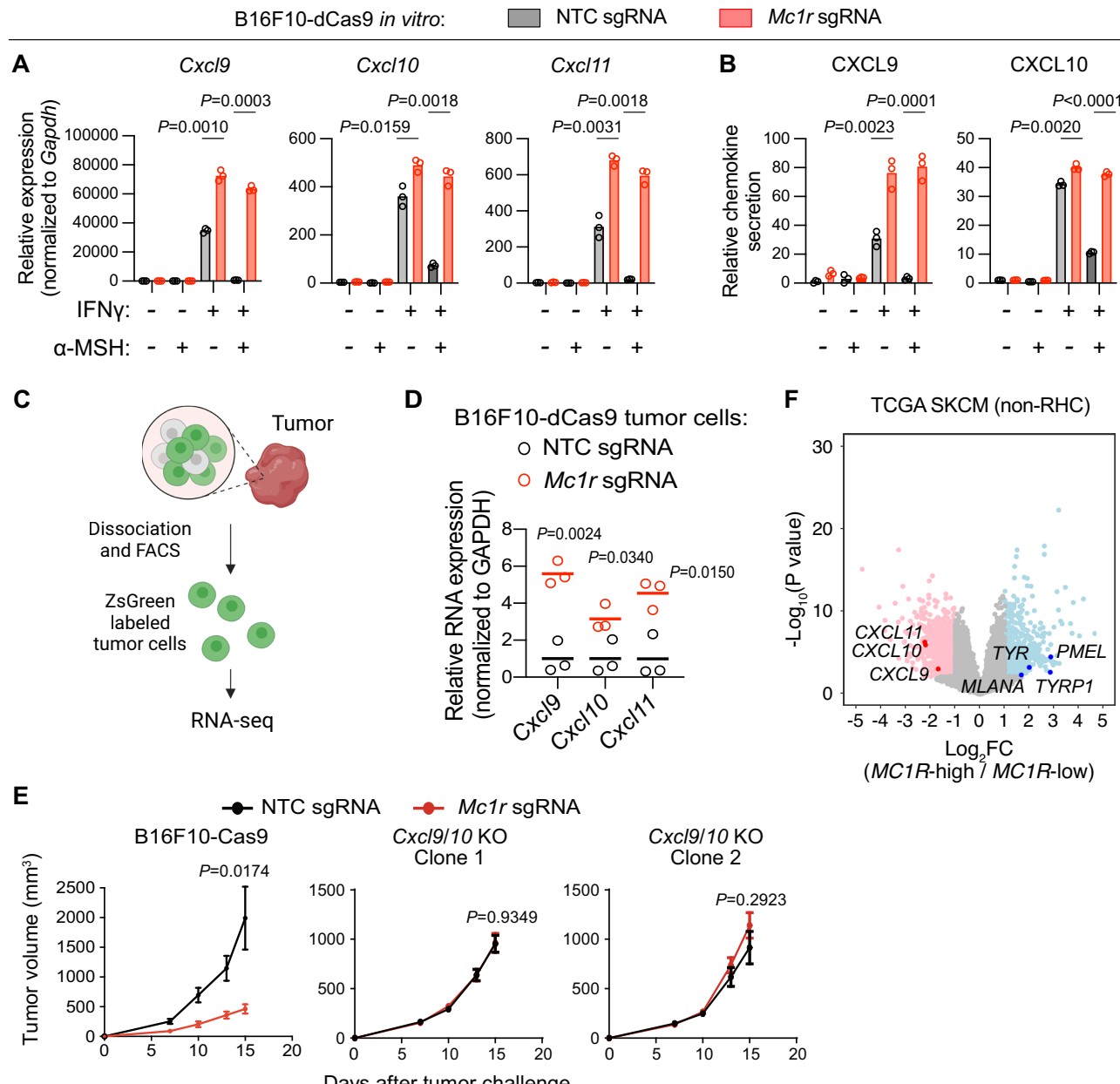

**Fig. 4 | B16F10 melanoma evades from immunosurveillance through MC1R-mediated repression of *Cxcl9/10* transcription. A** qPCR quantification of *Cxcl9/10/11* mRNA levels in B16F10-dCas9 cells transduced with NTC sgRNA or *Mc1r* sgRNA with indicated treatment of α-MSH (1 μM) and IFNγ (10 ng/ml) for 12 h. Lines indicate the mean of three independent samples. Welch's *t* tests (two-tailed) were used to determine statistical significance. **B** ELISA quantification of CXCL9 and CXCL10 secretion by B16F10-dCas9 cells transduced with NTC sgRNA or *Mc1r* sgRNA with indicated treatment of α-MSH (1 μM) and IFNγ (10 ng/ml) for 24 h. Lines indicate the mean of three independent samples. Student's *t* tests (two-tailed) were used to determine statistical significance. **C** Strategy to isolate zsGreen-labeled tumor cells for RNA-seq. Created with biorender.com. **D** RNA-seq quantification of *Cxcl9/10/11* expression in FACS-sorted zsGreen-positive tumor cells from B16F10-dCas9-zsGreen tumors expressing NTC sgRNA or *Mc1r* sgRNA. Wild-type C57/BL6 mice were used for tumor cell transplantation. Lines indicate the mean of three independent tumors; Welch's *t* tests (two-tailed) were used for statistical analysis.

**E** Effect of *Mc1r* deletion on the growth of tumors derived from B16F10-Cas9 cells versus tumors derived from two independent B16F10-Cas9 *Cxcl9/10* knockout (KO) clones. Wild-type C57/BL6 mice were used for tumor cell transplantation. Data are the mean ± s.e.m. (B16F10-Cas9: *n* = 8 for NTC sgRNA, 10 for *Mc1r* sgRNA; *Cxcl9/10* KO clone 1: *n* = 14 for NTC sgRNA, 13 for *Mc1r* sgRNA; *Cxcl9/10* KO clone 2: *n* = 13 for NTC sgRNA, 13 for *Mc1r* sgRNA). Welch's *t* tests (two-tailed) were used to determine statistical significance in tumor volumes. **F** Volcano plots depicting differentially expressed genes (computed with cBioPortal) in *MC1R*-high (*n* = 62) versus *MC1R*-low (*n* = 62) non-RHC melanomas in TCGA stratified by *MC1R* expression. Comparisons were made between the highest quartile versus the lowest quartile. Genes differentially expressed (*P* < 0.01) were labeled as pink (log₂FC < −1.5) or light sky-blue dots (log₂FC > 1.5). Genes indicating MC1R activation and chemokine repression were labeled respective as blue and red dots. *P* values were computed by Student's *t* test (two-tailed). Source data are provided as a Source Data file.

activation of MC1R by α-MSH stimulated the phosphorylation of CREB1 and ATF1. Expression of the *Mc1r* sgRNA blocked α-MSH-induced CREB1/ATF1 phosphorylation (Fig. 5C). By qPCR, we observed that IFNγ treatment resulted in robust induction of *Cxcl9/10/11* expression in HCmel1274-dCas9 cells transduced with the NTC sgRNA, which was significantly repressed when α-MSH was added simultaneously with IFNγ. In contrast, HCmel1274-dCas9 cells transduced with the *Mc1r* sgRNA expressed higher levels of *Cxcl9/10/11* upon IFNγ treatment, which could not be suppressed by α-MSH treatment (Fig. 5D).

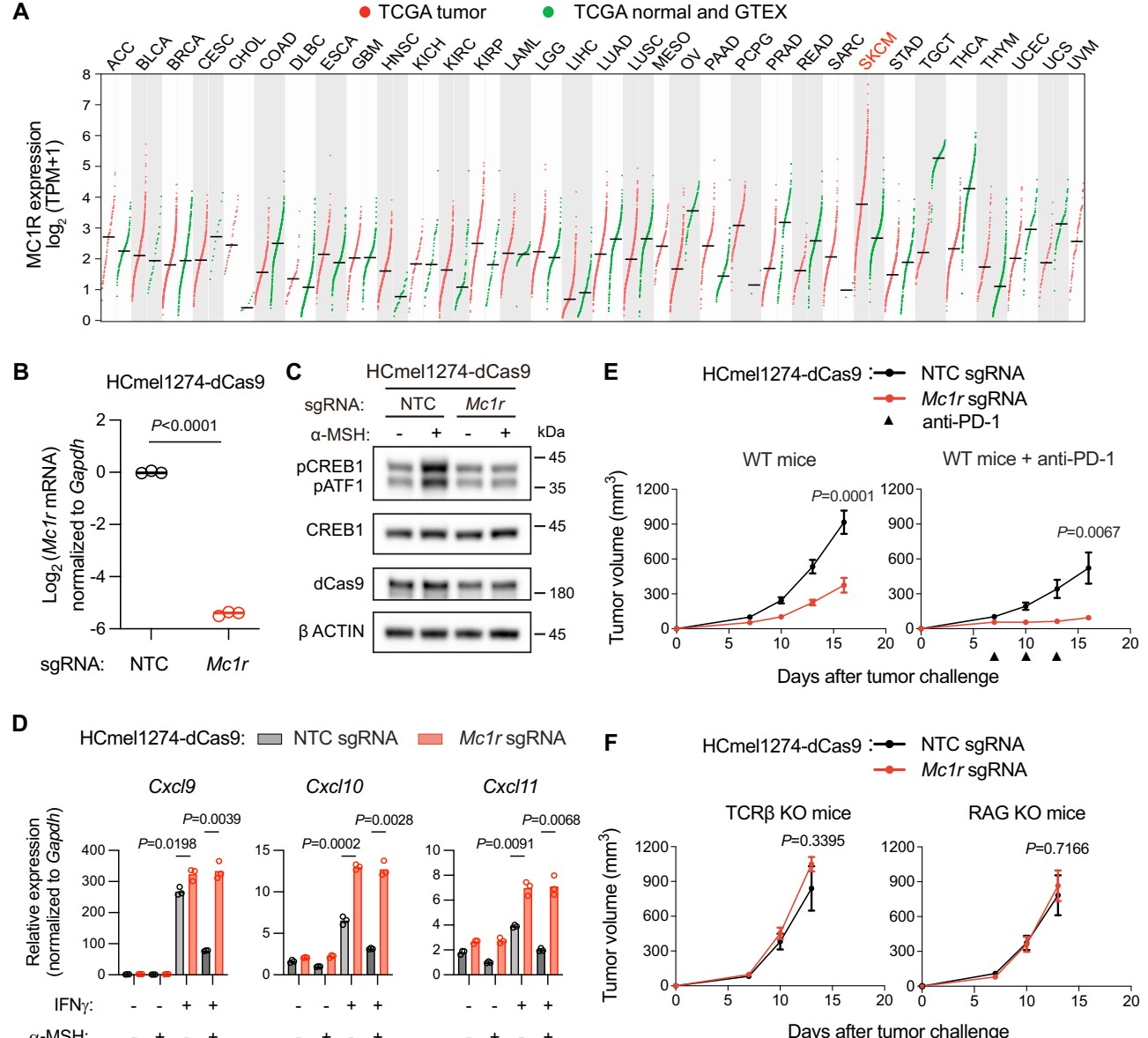

**Fig. 5 | MC1R promotes immune evasion of HCmel1274 melanoma. A** Expression levels of *MC1R*, assessed by log$_2$-transformed tag per million (TPM) reads, in tumor samples (*n* = 9664) versus normal tissues (*n* = 5540) based on TCGA and GTEX RNA-seq datasets. **B** qPCR quantification of *Mc1r* mRNA levels in HCmel1274-dCas9 cells transduced with NTC sgRNA or *Mc1r* sgRNA. Lines indicate the mean of three independent samples. The Welch's *t* test (two-tailed) was used to determine statistical significance. **C** Western blotting of the indicated proteins in HCmel1274-dCas9 cells transduced with NTC sgRNA or *Mc1r* sgRNA. Cells were untreated or treated with 1 µM α-MSH for 0.5 h. A representative result was shown from two independent experiments. Uncropped western blot images are provided as a Source Data file. **D** qPCR quantification of *Cxcl9/10/11* mRNA levels in HCmel1274-dCas9 cells transduced with NTC sgRNA or *Mc1r* sgRNA with indicated treatment of α-MSH (1 µM) and IFNγ (10 ng/ml) for 12 h. Lines indicate the mean of three

independent samples. Welch's *t* tests (two-tailed) were used to determine statistical significance. **E** Tumor volume of wild-type mice transplanted with HCmel1274-dCas9 tumors expressing NTC sgRNA or *Mc1r* sgRNA. Anti-PD-1 treatments are indicated as black triangles. Data are the mean ± s.e.m. (WT mice: *n* = 14 for NTC sgRNA, 15 for *Mc1r* sgRNA; WT mice+anti-PD-1: *n* = 15 for NTC sgRNA, 15 for *Mc1r* sgRNA). Welch's *t* tests (two-tailed) were used to determine the statistical significance of the differences in tumor volume. **F** Tumor volume of TCRβ KO or *Rag1* KO mice transplanted with HCmel1274-dCas9 tumors expressing NTC sgRNA or *Mc1r* sgRNA. Data are the mean ± s.e.m. (TCRβ KO mice: *n* = 6 for NTC sgRNA, 5 for *Mc1r* sgRNA; RAG KO mice: *n* = 7 for NTC sgRNA, 6 for *Mc1r* sgRNA). Welch's *t* tests (two-tailed) were used to determine the statistical significance of the differences in tumor volume. Source data are provided as a Source Data file.

To examine whether MC1R promotes immune evasion of HCmel1274, we transplanted HCmel1274-dCas9 cells transduced with NTC or *Mc1r* sgRNA into mice and monitored tumor growth. In wild-type mice, MC1R depletion slowed tumor growth and enhanced the efficacy of anti-PD-1 treatment (Fig. 5E). In contrast, MC1R depletion did not affect tumor growth in immunodeficient TCRβ or RAG1 knockout mice (Fig. 5F). Together with our findings in B16F10, these results suggest that MC1R could be an immunotherapy target for melanocytic melanoma.

## GNAS activation promotes immune evasion of B16F10 melanoma

As MC1R is known to activate the GNAS-containing heterotrimeric G protein[38], we wondered whether GNAS activation was sufficient to restore immune evasion of MC1R-depleted B16F10 melanoma. The R201C mutation has been shown to constitutively activate GNAS[25,50]. We thus expressed the wild-type or the R201C mutant forms of GNAS in MC1R-depleted B16F10-dCas9 cells (Supplementary Fig. S8A). We

observed higher CRE-Fluc reporter activity in cells expressing GNAS R201C relative to wild-type GNAS (Supplementary Fig. S8B), confirming the constitutive activity of the R201C mutant. In addition, GNAS R201C expression reduced IFNγ-induced *Cxcl9/10/11* expression and CXCL9/10 secretion compared to wild-type GNAS (Fig. 6A and Supplementary Fig. S8C) in MC1R-depleted B16F10-dCas9 cells, indicating that GNAS acts downstream of MC1R in these cells.

We then transplanted MC1R-depleted B16F10-dCas9 cells expressing wild-type GNAS or the R201C mutant into mice and monitored tumor growth over time. In wild-type mice, tumors expressing the GNAS R201C mutant grew faster than those expressing wild-type GNAS, resulting in shorter host survival time. In contrast, when transplanting tumor cells expressing the wild-type or the R201C mutant forms of GNAS into the NCG mice, we did not observe significant difference in tumor growth or host survival time (Fig. 6B and Supplementary Fig. S8D). Thus, GNAS activation was sufficient to restore immune evasion of MC1R-depleted B16F10 melanoma.

We further analyzed infiltrating immune cells in these two groups of tumors grown in wild-type mice by flow cytometry. The GNAS R201C mutant-expressing tumors showed decreased numbers of CD8⁺ and CD4⁺ T cells, as well as NK cells in comparison to tumors expressing wild-type GNAS (Fig. 6C). Moreover, we observed significant lower percentages of IFNγ- and GZMB-positive CD8⁺ T cells from GNAS R201C mutant-expressing tumors, which were accompanied by decreased MFI of these two molecules (Supplementary Fig. S8E). Taken together, these results reveal that GNAS activation represses *Cxcl9/10/11* transcription to impair T cell infiltration and effector functions, thus enabling B16F10 melanoma to escape immunosurveillance.

### PKA-CREB mediates immune evasion of B16F10 melanoma

GNAS activation led to the production of the second messenger cAMP, which in turn can activate several effectors, including PKA, EPAC (a guanine-nucleotide-exchange factor), and cyclic-nucleotide-gated ion channels[51]. We tested the requirement of the classical cAMP effector PKA for MC1R-mediated immune evasion of B16F10 melanoma. Two paralogs of PKA encoded by *Prkaca* and *Prkacb* are present in the mouse genome. We therefore adopted an optimized double sgRNA vector system[52] to individually or simultaneously delete the two genes from B16F10 cells. In B16F10-Cas9 cells transduced with two NTC sgRNAs, α-MSH could stimulate CREB1/ATF1 phosphorylation and repress IFNγ-induced *Cxcl9/10/11* expression. Similar observations were made in B16F10-Cas9 cells transduced with single sgRNA targeting *Prkaca* or *Prkacb*. In contrast, in B16F10-Cas9 cells transduced with two sgRNAs targeting both *Prkaca* and *Prkacb*, α-MSH-mediated CREB1/ATF1 phosphorylation and *Cxcl9/10/11* repression was impaired (Fig. 6D and Supplementary Fig. S9A). These results reveal functional redundancy between these two PKA paralogs in mediating MC1R signaling in B16F10 cells.

We next transplanted B16F10-Cas9 cell populations expressing two NTC sgRNAs, single sgRNA targeting *Prkaca* or *Prkacb*, or two sgRNAs targeting *Prkaca* and *Prkacb* into mice and monitored tumor growth over time. In wild-type mice, loss of both *Prkaca* and *Prkacb* led to slower tumor growth and significant extension of host survival in comparison to tumors with two NTC sgRNAs, or single sgRNA targeting *Prkaca* or *Prkacb* (Fig. 6E and Supplementary Fig. S9B). In contrast, when transplanting B16F10-Cas9 cells expressing two NTC sgRNAs or two sgRNAs targeting *Prkaca* and *Prkacb* into the NCG mice, we did not observe significant difference in tumor growth or host survival time (Fig. 6E and Supplementary Fig. S9C). Reintroduction of an sgRNA-resistant *Prkaca* cDNA fully rescued the growth of *Prkaca/Prkacb* double knockout tumors (Supplementary Fig. S9D, E). Moreover, *Prkaca/Prkacb* double deletion in B16F10-Cas9 tumors resulted in increased T cell infiltration and enhanced effector functions of infiltrating CD8⁺ T cells in comparison to control (Supplementary Fig. S9F,

G). These results indicate that PKA functions downstream of MC1R to promote immune evasion of B16F10 melanoma.

Activated PKA can phosphorylate a variety of substrates such as the CREB family proteins, which are signal inducible basic region/leucine zipper (bZIP) transcription factors, inducing CREB, cAMP responsive element regulatory protein (CREM) and ATF1[52]. To examine whether CREB proteins function downstream of PKA to promote immune evasion, we expressed a dominant negative construct A-CREB in B16F10 cells (Supplementary Fig. S9H). A-CREB is composed of an acidic amphipathic extension fused to the N terminus of the CREB's leucine zipper domain. Upon dimerization with wild-type CREB, the acidic extension of A-CREB interacts with the basic region of wild-type CREB to prevent the latter from binding to DNA[53]. B16F10 cells transduced with A-CREB expressed higher levels of *Cxcl9/10/11* in response to IFNγ relative to cells transduced with a control vector. In addition, the repression of *Cxcl9/10/11* expression by MC1R activation (α-MSH treatment) was partially relieved by A-CREB expression (Fig. 6F). We next transplanted B16F10 cells expressing the vector control or A-CREB into mice and observed that A-CREB expression significantly slowed tumor growth in wild-type mice. Moreover, the antitumor effect of A-CREB was enhanced by anti-PD-1 treatment (Fig. 6G). We thus conclude that CREB is required for B16F10 melanoma cells to evade immunosurveillance.

### Oncogenic activation of GNAS represses *CXCL9/10/11* expression in human cancers

Although MC1R activation is a melanoma-specific mechanism of immune evasion, hotspot mutations constitutively activating GNAS are found in diverse cancer types (Fig. 7A). We thus hypothesized that activation of GNAS could be a general strategy employed by human cancers to inhibit T cell infiltration. To examine whether constitutive activation of GNAS correlates with reduced *CXCL9/10/11* expression in human cancers, we focused on pancreatic adenocarcinoma (PAAD) and liver hepatocellular carcinoma (LIHC), as these two cancer types contained more than 3 cases with *GNAS* R201C/H/L mutations occurring at greater than 1% of patients (Fig. 7A). We compared the transcriptomes of *GNAS* R201C/H/L mutant cases versus *GNAS* wild-type cases of PAAD and LIHC. The expression levels of *CXCL9/10/11* were significantly lower in the *GNAS* mutant group compared with the *GNAS* wild-type group (Fig. 7B, C).

GNAS activation results in the activation of adenylyl cyclase, raising intracellular cAMP levels to activate PKA. Forskolin (FSK) is a diterpene natural product that activates adenylyl cyclase independent of GNAS activation[54]. We treated a panel human cancer cell lines of diverse origins and two patient-derived liver tumor organoids with IFNγ alone or IFNγ plus FSK. We found that the majority of these cell line and organoid models responded to FSK by downregulating IFNγ-induced *CXCL9/10/11* expression (Fig. 7D). These results demonstrate that GNAS-PKA signaling can repress *CXCL9/10/11* transcription in diverse cancer types.

### Oncogenic activation of GNAS promotes immune evasion of mouse liver and breast tumors

The observed correlation between *GNAS* mutations with reduced *CXCL9/10/11* expression in human cancers prompted us to examine the effect of GNAS R201C in mouse models of tumor types other than melanoma. We first derived a liver cancer cell line MAP (*Myc*^OE, *Apc*⁻/⁻, *Tp53*⁻/⁻) from mouse liver tumors induced by *Myc* overexpression and *Apc/Tp53* deletion (Fig. 7E and Supplementary Fig. S10A) using the hydrodynamic tail-vein injection method[55]. In MAP cells, expression of the GNAS R201C mutant reduced IFNγ-induced *Cxcl9/10/11* expression compared to expression of wild-type GNAS (Fig. 7F). When transplanted into wild-type mice, MAP tumors expressing GNAS R201C grew significantly faster than those expressing wild-type GNAS. In the presence of anti-PD-1 treatment, the effect of the GNAS R201C mutant

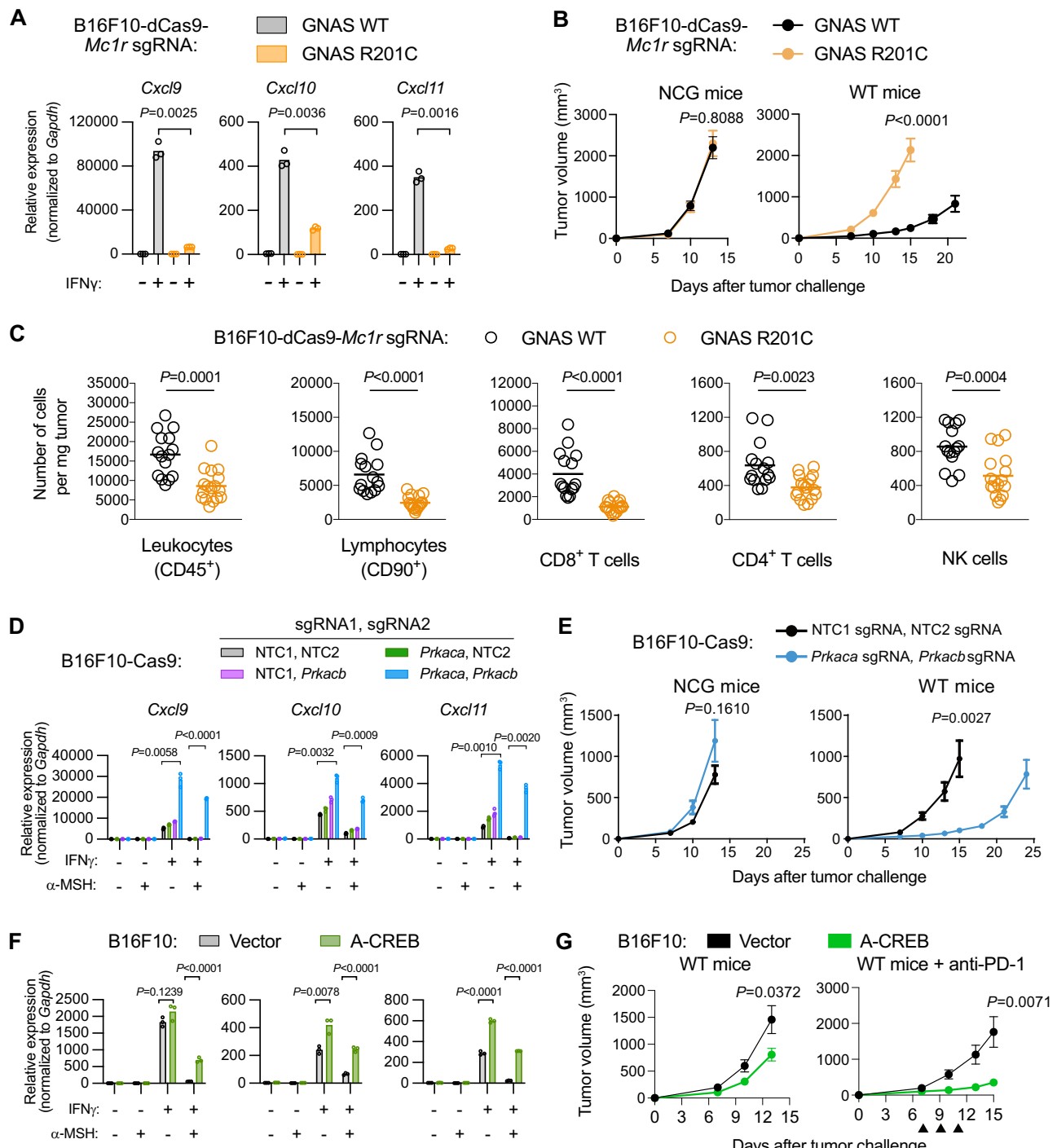

**Fig. 6 | GNAS-PKA-CREB signaling promotes immune evasion of B16F10 melanoma. A** qPCR quantification of *Cxcl9/10/11* mRNA levels in B16F10-dCas9-*Mc1r* sgRNA cells expressing wild-type GNAS or the R201C mutant of GNAS. Cells were untreated or treated with IFNγ (10 ng/ml) for 12 h as indicated. Lines indicate the mean of three independent samples. Welch's *t* tests (two-tailed) were used to determine statistical significance. **B** Growth of B16F10-dCas9-*Mc1r* sgRNA tumors expressing wild-type GNAS or the R201C mutant of GNAS in NCG and wild-type mice. Data are the mean ± s.e.m. (NCG mice: *n* = 10 per group; WT mice: *n* = 11 per group). Welch's *t* tests (two-tailed) were used to determine statistical significance. **C** Number of tumor-infiltrating immune cell subsets per mg tumor analyzed by flow cytometry comparing B16F10-dCas9-Mc1r sgRNA tumors expressing wild-type GNAS (*n* = 15) versus GNAS R201C mutant (*n* = 17). Lines indicate the mean. Welch's *t* tests (two-tailed) were used to determine statistical significance. **D** qPCR quantification of *Cxcl9/10/11* mRNA levels in B16F10-Cas9 cells transduced with indicated combinations of sgRNAs targeting protein kinase A (encoded by *Prkaca* and

*Prkacb*). Cells were treated with α-MSH (1 μM) and IFNγ (10 ng/ml) for 12 h as indicated. Lines indicate the mean of three independent samples. Welch's *t* tests (two-tailed) were used to determine statistical significance. **E** Growth of B16F10-Cas9 tumors expressing NTC1/NTC2 sgRNAs versus *Prkaca/Prkacb* sgRNAs in NCG and wild-type mice. Data are the mean ± s.e.m. (NCG mice: *n* = 10 per group; WT mice: *n* = 11 per group). Welch's *t* tests (two-tailed) were used to determine statistical significance. **F** qPCR quantification of *Cxcl9/10/11* mRNA levels in B16F10 cells expressing vector or A-CREB. Cells were treated with α-MSH (1 μM) and IFNγ (10 ng/ml) for 12 h as indicated. Lines indicate the mean of three independent samples. Welch's *t* tests (two-tailed) were used to determine statistical significance. **G** Growth of B16F10 tumors expressing vector or A-CREB in wild-type mice. Data are the mean ± s.e.m. (WT mice: *n* = 12 for vector, 13 for A-CREB; WT mice+anti-PD-1: *n* = 12 for vector, 13 for A-CREB). Anti-PD-1 treatments are indicated as black triangles. Welch's *t* tests (two-tailed) were used to determine statistical significance. Source data are provided as a Source Data file.

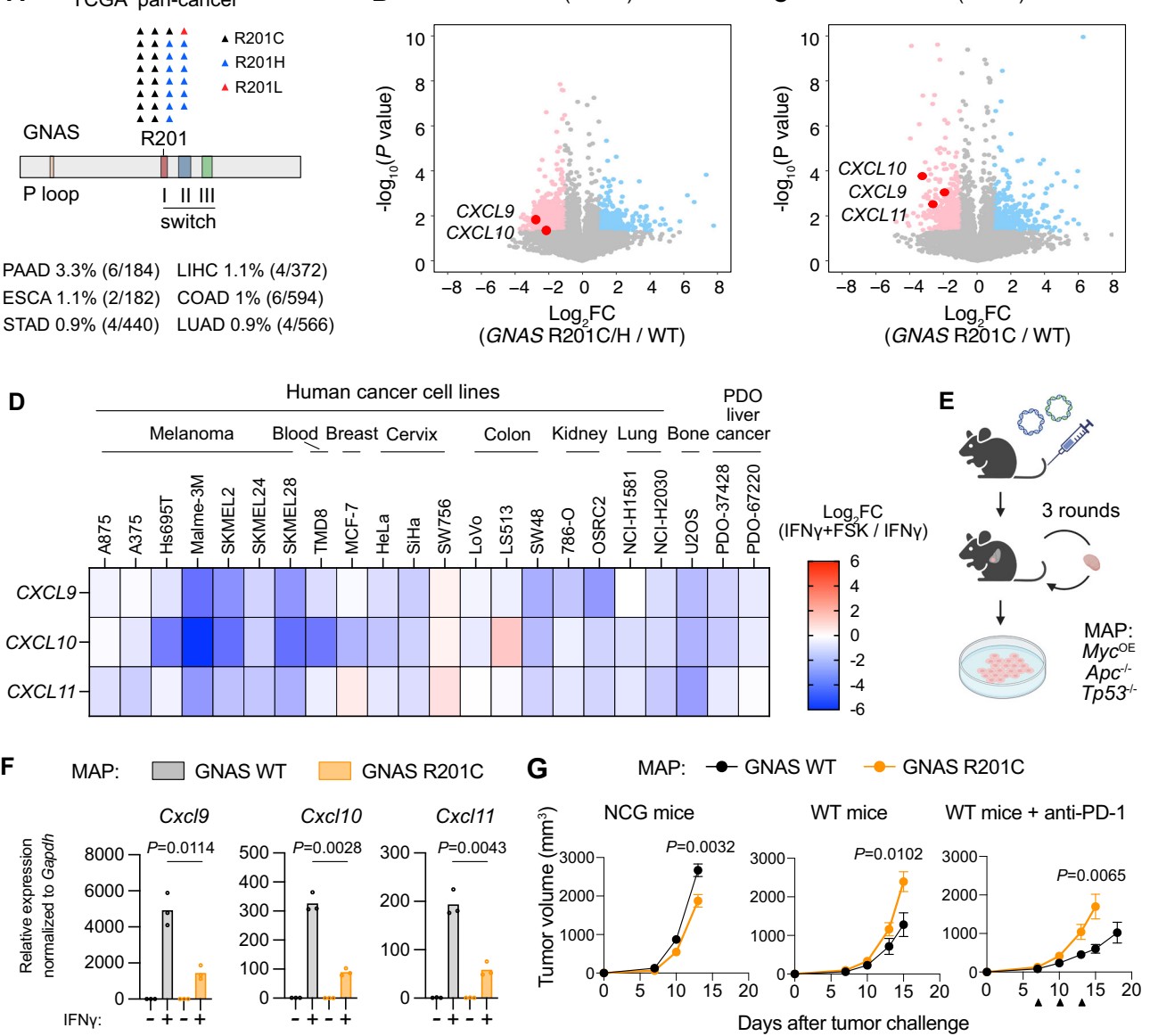

**Fig. 7 | GNAS activation represses *CXCL9/10/11* expression and promote immune evasion across diverse cancer types. A** Number and frequency of cases with *GNAS* R201C/H/L mutations in TCGA Pan-Cancer patient cohorts. **B**, **C** Volcano plots depicting differentially expressed genes (computed with cBioPortal) in *GNAS* R201C/H-mutant (*n* = 6) versus *GNAS* wild-type (*n* = 166) pancreatic adenocarcinoma (PAAD) (**B**) or in *GNAS* R201C/H-mutant (*n* = 4) versus *GNAS* wild-type (*n* = 357) Liver hepatocellular carcinoma (LIHC) (**C**) in TCGA. Genes differentially expressed (*P* < 0.05) were labeled as pink (log₂FC < −1) or light sky-blue dots (log₂FC > 1). Genes indicating chemokine repression were labeled as red dots. *P* values were computed by Student's *t* test (two-tailed). **D** Heatmap depicting the effect of forskolin (FSK) on IFNγ-induced expression of *CXCL9/10/11* in indicated human cell lines and patient-derived liver tumor organoids (data were not replicated). Cells and organoids were treated with FSK (50 μM) and IFNγ (10 ng/ml) for 12 h as indicated before qPCR measurements. ACTB mRNA was used for data normalization. **E** Strategy to derive the MAP liver cancer cell line. Liver tumors were induced by hydrodynamic tail-vein injection of plasmids encoding the piggyBac transposase, Myc, Cas9 and sgRNAs targeting *Apc* and *Tp53*. After tumor formation, the resulting tumor blocks were subcutaneously transplanted into wild-type mice. The MAP cell line was derived from dissociated tumors after three rounds of transplantation. Created with biorender.com. **F** qPCR quantification of *Cxcl9/10/11* mRNA levels in MAP liver cancer cells expressing wild-type GNAS or the R201C mutant of GNAS. Cells were untreated or treated with IFNγ (10 ng/ml) for 12 h. Lines indicate the mean of three independent samples. Welch's *t* tests (two-tailed) were used to determine statistical significance. **G** Growth of MAP tumors expressing wild-type GNAS or the R201C mutant of GNAS in NCG and wild-type mice. Data are the mean ± s.e.m. (NCG mice: *n* = 10 per group; WT mice: *n* = 14 for GNAS WT, 13 for GNAS R201C; WT mice+anti-PD-1: *n* = 12 per group). Anti-PD-1 treatments are indicated as black triangles. Welch's *t* tests (two-tailed) were used to determine statistical significance. Source data are provided as a Source Data file.

in promoting MAP tumor growth was even more pronounced. In contrast, expression of the GNAS R201C mutant did not lead to faster tumor growth in comparison to expression of wild-type GNAS in NCG mice (Fig. 7F). Thus, GNAS activation promoted immune evasion of MAP tumors.

We next tested whether GNAS activation could promote immune evasion of 4T1-derived breast cancer in mice. 4T1 tumors exhibit limited immune infiltration, whereas ectopic expression of hemagglutinin A (HA) was reported to enhance immune surveillance in vivo[56]. Similar to observations made from B16F10 and MAP tumors, expression of the GNAS R201C mutant in 4T1-HA repressed *Cxcl9/10/11* expression and promoted tumor growth in comparison to expression of wild-type GNAS (Supplementary Fig. S10B, C). The pro-tumor effect of GNAS R201C in 4T1-HA compared to wild-type GNAS was not

observed in TCRβ knockout mice (Supplementary Fig. S10C). Taken together, activation of GNAS by the oncogenic R201C mutation led to immune evasion in mouse models of liver and breast cancers.

## Discussion

In this study, we used in vivo CRISPR screening, which is a powerful approach for discovering genes that promote immune evasion as shown by several studies in recent years[32,57–60]. These prior studies have focused on highly expressed genes or epigenetic regulators, discovering mechanisms that maintain tumor cells in a state that is resistance to immune attack. However, these studies have not been able to reveal the upstream signals that instruct tumor cells to escape immunosurveillance. Motivated by the need to fill this knowledge gap, we focused our screening effort on membrane and secreted proteins responsible for inter-cellular communications, resulting in the discovery of MC1R.

MC1R is a cAMP-stimulating GPCR regulating skin pigmentation in mammals[37]. The activation of MC1R stimulates the production of eumelanin, which has strong shielding capacity against ultraviolet (UV) radiation[61]. Inactivating polymorphisms in the *MC1R* gene are associated with increased melanoma risk likely due to UV-induced DNA damage[62]. Our study reveals a distinct function of MC1R in promoting the escape of a subset of melanomas with melanocytic features from immune surveillance. By transcriptionally repressing IFNγ-induced *CXCL9*, *CXCL10*, and *CXCL11* expression, MC1R activation impedes the recruitment of CXCR3-positive T cells into the TME. Such anti- versus pro-melanoma functions of MC1R may reflect different selection pressures during melanoma initiation versus progression. Neoplastic transformation of melanocytes is facilitated by UV-induced DNA damage, which at the same time increases the antigenicity of melanoma cells. To escape immune surveillance at later stages of melanoma progression, melanoma cells hijack MC1R signaling to repress T cell chemoattraction.

MC1R is activated upon binding to α-MSH, which is an endogenous peptide hormone of the melanocortin family[63]. A recent study revealed that subcutaneous implantation of tumors induced pituitary α-MSH production in mice and that serum α-MSH concentration was elevated in cancer patients[64]. Together with our results, it is plausible that melanoma cells with high *MC1R* expression can take advantage of elevated circulating α-MSH to activate MC1R signaling to dampen antitumor T cell response. MC1R activity is naturally antagonized by agouti signaling protein (ASIP)[65]. A previous study showed that expression of ASIP in B16F10 limited its lung metastasis[66]. It is thus possible to develop antagonists of MC1R to test whether therapeutic blockade of MC1R activation can augment antitumor immunity in melanoma patients.

Our genetic dissection of the pathway downstream of MC1R revealed the role of GNAS-PKA signaling in promoting immune evasion. GNAS is the most frequently mutated heterotrimeric G protein associated with malignancies arising from the pituitary gland, colon, pancreas, kidney, and liver[20]. While previous studies have revealed that oncogenic GNAS mutants promote hyperplastic growth and tumorigenesis[67–69], our study demonstrates that they also promote immune evasion by repressing *CXCL9/10/11* across human cancers. In addition to oncogenic activation of GNAS, a variety of cancer-associated alterations can enhance GPCR-GNAS-PKA signaling, including activating mutations in the catalytic subunit of PKA, inactivating mutations in the regulatory subunit of PKA, and overexpression of cAMP-stimulating GPCRs[70–73]. Thus, counteracting aberrant activation of cAMP-PKA signaling in cancer cells harboring these alterations may activate antitumor T cell response and improve the efficacy of ICB therapies.

Among hundreds of IFNγ-induced genes, *CXCL9/10/11* and several MHC class II genes were particularly sensitive to GPCR-GNAS-PKA signaling. ATAC-seq revealed reduced chromatin accessibility of promoter and enhancer elements at genomic loci harboring these genes. Motif analysis predicted enrichment of sequences recognized by the ETS family of transcription factors at these cis-regulatory elements. Moreover, expression of the dominant negative A-CREB construct attenuated the repressive effect of GPCR-GNAS-PKA signaling on *CXCL9/10/11* transcription, suggesting the involvement of CREB. How CREB represses the activation of these cis-regulatory elements, possibly by interfering with ETS activity, remain to be investigated.

GPCR-GNAS-PKA signaling is activated in many tissues under physiological conditions[74]. Given the connection of cAMP-PKA signaling to antitumor immunity reported in this study, it will be interesting to examine whether physiological cAMP-PKA signaling protects normal tissues against autoimmunity through similar mechanisms.

## Methods

The research in this study complies with the animal welfare guidelines and was approved by the NIBS Animal Use and Care Committee.

### Mice

Wild-type C57BL/6 mice and BALB/c mice were obtained from the Transgenic Research Center at NIBS or were purchased from Beijing Vital River Laboratory Animal Technology (Beijing, China). NCG (NOD/ShiLtJGpt-*Prkdc*em26Cd52*Il2rg*em26Cd22/Gpt) mice were purchased from GemPharmatech (Nanjing, Jiangsu, China). TCRβ knockout mice (JAX stock# 002118), *Rag1* knockout mice (JAX stock#002216), and Cas9 transgenic mice (JAX stock# 026179) were bred in house. All mice were housed in the specific-pathogen free animal facility at NIBS under a 12-h light–dark cycle with free access to food and water, 23–25 °C and 50–56% humidity. All animal experiments were approved by the NIBS Animal Use and Care Committee.

### Cell lines and patient-derived organoids

B16F10, MCF-7, SiHa, A875, and A375 were purchased from Cell Resource Center, Peking Union Medical College (Beijing, China). HCmel1274 was purchased from the American Type Culture Collection (ATCC). LS513 was purchased from Meisen Cell, Zhejiang, China. SKMEL2, SKMEL28, SKMEL24, Hs695T, Malme-3M, C32, SW756, 786-O, and OSRC2 were purchased from Cobioer, Nanjing, China. 4T1 was a gift from Dr. Xiaodong Wang (NIBS, Beijing, China). TMD8 was a gift from BeiGene Co., Ltd., Beijing, China. HEK293T, HeLa, LoVo, SW48, NCI-H1581, NCI-H2030, and U2OS were gifts from Dr. Deepak Nijhawan (University of Texas Southwestern Medical Center, Dallas, Texas, USA). MAP was derived in house. B16F10, HCmel1274, 4T1, LoVo, LS513, SW48, 786-O, OSCRC2, NCI-H1581, and NCI-H2030 cells were cultured in RPMI-1640 supplemented with 10% fetal bovine serum, 2 mM L-glutamine, and 1% penicillin–streptomycin. HEK293T, MAP, MCF-7, HeLa, SiHa, SW756, A875, A375, and U2OS cells were cultured in DMEM supplemented with 10% fetal bovine serum, 2 mM L-glutamine, and 1% penicillin–streptomycin. SKMEL2, SKMEL28, SKMEL24, HS695T and C32 cells were cultured in MEM supplemented with 10% fetal bovine serum, 1% nonessential amino acids, 1 mM sodium pyruvate, 2 mM L-glutamine, and 1% penicillin–streptomycin. Malme-3M cells were cultured in IMDM supplemented with 20% fetal bovine, 2 mM L-glutamine, and 1% penicillin–streptomycin. All cell lines were grown in tissue culture incubators at 37 °C and 5% CO₂ and confirmed to be mycoplasma-free on a weekly basis using a PCR-based assay. Patient-derived liver tumor organoids were provided by K2 Oncology (Beijing, China).

### Lentivirus production

A total of $0.6 \times 10^6$ HEK293T cells were plated per well in sic-well plates. The following day, 1 μg of plasmids containing the corresponding lentiviral vector, psPAX2 (addgene #12260), and pMD2.G (addgene #12259) at a mass ratio of 5:3:2 were transfected into HEK293T cells using the polyethylenimine (PEI) transfection reagent. Between 48 and

72 h post transfection, virus-containing cell culture supernatant was passed through 0.45-micron filter and stored at −80 °C for later use.

## Construction of sgRNA libraries

A list of 6053 genes encoding mouse membrane or secreted proteins was identified by homology searches with genes encoding human membrane or secreted proteins[33]. Four sgRNA sequences per gene were obtained from the Brie library[75]. The resulting sgRNA sequences were divided into three sub-libraries: sub-libraries A and B both contain 8072 sgRNAs targeting 2018 genes and 500 nontargeting control sgRNAs. Sublibrary C contains 8068 sgRNAs targeting 2017 genes and 504 nontargeting control sgRNAs. Three pools of sgRNA oligos were synthesized by Twist Bioscience (South San Francisco, California, USA), cloned into the lentiGuide-Puro vector (#52963; Addgene, Watertown, Massachusetts, USA) and packaged into a lentivirus as previously described[76].

## In vivo CRISPR screening in B16F10 melanoma

B16F10 cells were transduced with SpCas9 (lentiCas9-Blast, #52962; Addgene, Watertown, Massachusetts, USA) via lentiviral infection. In total, $10^7$ B16F10-Cas9 cells were infected with each lentiviral library at a multiplicity of infection (MOI) of ~0.3. Forty-eight hours after infection, the cells were reseeded and selected with 2 μg/ml puromycin. Fourteen days after cell growth in vitro, $10^7$ sgRNA-transduced B16F10-Cas9 cells were used as an in vitro control. For in vivo growth, $2 \times 10^6$ sgRNA-transduced B16F10-Cas9 cells were subcutaneously injected into one dorsal flank of wild-type female C57BL/6 mice (6–8 weeks old, $n = 5$ animals, both flanks were injected). Tumors were dissected from sacrificed animals after fourteen days of in vivo growth. Between 6-8 tumors (excluding outliers based on tumor weight) were included in the analysis. Genomic DNA was extracted as previously described[77]. For the first step of sgRNA amplification, fifteen 25-μl PCRs (each containing 1 μg of genomic DNA) were performed using NEBNext Ultra II Q5 master mix (NEB, Ipswich, Massachusetts, USA) with a forward primer (5′-CCTACAC GACGCTCTTCCGATCTNNNNNNNNNNNNNNNNNNNNN-GCTTTATAT ATCTTGTGGAAAGGACGAAACACC-3′), a reverse primer (5′-CAGA CGTG-TGCTCTTCCGATC-TCCGACTCGGTGCCACTTTTTCAA-3′) and the following cycling program: 98 °C for 5 min; 22 cycles of 98 °C for 10 s, 69 °C for 30 s, and 65 °C for 45 s; and a final extension at 65 °C for 5 min. The first-step PCR products were recovered with PCR-cleanup columns and used as DNA templates for five 25-μl PCRs (each containing 6 ng of purified first-step PCR products) with a forward primer (5′-AATGATACGGCGACCACCGAGATCTACACTCTTTCCCT ACACGACGCTCTTCCGATCT-3′), indexed reverse primers (5′-CAAGC AGAAGACGGCATACGAGAT-8-nucleotide-index-GTGACTGGAGTTC AGACG-TGTGCTCTTCCGATCT-3′) and the following cycling program: 98 °C for 5 min; 10 cycles of 98 °C for 10 s, 69 °C for 30 s, and 65 °C for 45 s; and a final extension at 65 °C for 5 min. The second-step PCR products were separated by a 2% agarose gel, purified and sequenced by Illumina HiSeq PE150 (Novogene, Beijing, China). CRISPR screening data were analyzed by MAGeCK (v0.5.9.2) as previously described[34,77].

## Cell line engineering

To deplete MC1R, B16F10 and HCmel1274 cells were first transduced with nuclease-dead Cas9 fused with KRAB and then transduced with the following *Mc1r* sgRNA, 5′-CCTTTCCCTCGCCAAGCCA-3′. To generate knockout cell lines, B16F10 cells were first transduced with Cas9 and then transduced with the following sgRNAs: *Mc1r* sgRNA, 5′-AGA GCACGACGGGCTGTCGT-3′; *Prkaca* sgRNA: 5′-CTAAGATCTTCATG GCGTAG-3′; *Prkacb* sgRNA, 5′-ATCCCAGGGT-TACAATAAGG-3′; *Cxcl9* sgRNA, 5′-ATCGTGCATTCCTTATCACT-3′; *Cxcl10* sgRNA, 5′-ACTCAC ATGATCTCAACACG-3′; *Gnas* sgRNA, 5′-CTGGTCACTTGGCACGTAGT-3′. As controls, the following nontargeting sgRNAs were used: NTC sgRNA, 5′-ACCCACGTATGTACTCGGGA-3′; NTC1 sgRNA, 5′-CGAG-GCT TAACGCCAGATTC-3′; NTC2 sgRNA, 5′-CAGTGCTAACCTTGCATTG-3′; mouse *Rosa26* sgRNA, 5′- GCATTCTACACGTTATTGC-3′; mouse *H11* sgRNA, 5′-AACACTAGTGCACTTATCC-3′;. For single sgRNA delivery, the lentiGuide-Puro vector (Addgene #52963) was used; for two sgRNAs delivery, an optimized vector encoding two sgRNAs driven by human and mouse U6 promoters was used[52]. Stable expression of GNAS in B16F10, MAP, and 4T1-HA was achieved by lentiviral transduction of GNAS cDNA driven by HSV-TK, EF1α, and SFFV promoters, respectively. Stable expression of MC1R or PRKACA in B16F10 were achieved by lentiviral transduction of MC1R cDNA or PRKACA cDNA driven by the HSV-TK promoter. Stable expression of CRE-luciferase reporter in B16F10 was also achieved by lentiviral infection. Stable cell lines were selected by antibiotic resistance markers encoded on the lentiviral vectors.

## Tumor challenge

For in vivo tumor challenge experiments, all animals were 6–10-week-old female mice. A maximal tumor length of 2 cm approved by the NIBS Animal Use and Care Committee was not exceeded. For the B16F10 tumor model, $2 \times 10^6$ cells were subcutaneously injected into wild-type C57BL/6 mice and Cas9 transgenic mice, and $0.5 \times 10^6$ cells were subcutaneously injected into TCRβ knockout mice and NCG mice. For the HCmel1274 tumor model, $1 \times 10^6$ cells were subcutaneously injected into wild-type C57BL/6 mice, TCRβ knockout mice and *Rag1* knockout mice. For the MAP ($Myc^{OE}$, $Apc^{-/-}$, $Tp53^{-/-}$) tumor model, $2 \times 10^6$ cells were subcutaneously injected into wild-type C57BL/6 and NCG mice. For the 4T1 tumor model, $0.5 \times 10^6$ cells were subcutaneously injected into wild-type BALB/c mice or TCRβ knockout mice. For tumor challenge experiments combined with anti-PD-1 therapy, 100 μg of anti-mouse PD-1 antibody for B16F10 and MAP tumors and 200 μg of anti-mouse PD-1 antibody for HCmel1274 tumor were intraperitoneally injected into each mouse at day 7, 10, and 13 after tumor challenge. Tumor length (L) and width (W) were measured by a Vernier caliper at the indicated times, and tumor volumes were calculated by LxW$^2$x3.14/6. For mouse survival curve analysis, end points were defined as when the tumor length reached 2 cm.

## Immunostaining of tumor-infiltrating CD8 + T cells

In total, $0.8 \times 10^6$ tumor cells were subcutaneously injected into C57BL/6 mice. After 14 days, tumor tissues were dissected, fixed in 4% paraformaldehyde for 24 h, and dehydrated in 30% sucrose for 48 h. For tissue section staining, tissues were embedded in Optimal Cutting Temperature (OCT) compound, frozen, and cryo-sectioned (15–20 microns). Sections were permeabilized for 30 min in PBS with 0.1% Triton X-100 (PBST) and blocked for 1 h in PBS with 3% BSA. To stain for CD8$^+$ T cells, rabbit anti-mouse CD8a (Abcam, ab217344, clone: EPR21769, dilution: 1:500) was used as the primary antibody, and AF647-donkey anti-rabbit IgG (Jackson Immuno Research, 711-605-152, clone: 30-F11, dilution: 1:500) was used as the secondary antibody. Stained tissue sections were imaged on a Zeiss LSM800 confocal microscope. Microscopy data were analyzed using OlyVIA software (OLYMPUS).

## Analysis of tumor-infiltrating immune cells by flow cytometry

In total, $2 \times 10^6$ tumor cells were subcutaneously injected into C57BL/6 mice. After 12–14 days, tumor tissues were dissected out and then minced and dissociated in RPMI-1640 containing collagenase (1 mg/ml collagenase I; Roche, Basel, Switzerland) and DNase I (200 μg/ml; Sigma, St. Louis, Missouri, USA) with constant stirring at 37 °C for 20 min. Single cells were obtained by filtering through a 100-micron filter. Immune cells were stained with the following monoclonal antibodies: anti-mouse CD45-APC/Cyanine7 (Biolegend, 103116, clone: 30-F11, dilution: 1:400), anti-mouse CD90.2-PerCP/Cyanine5.5 (Biolegend, 105338, clone: 30-H12, dilution: 1:400), anti-mouse CD4-BV421

(Biolegend, 100438, clone: GK1.5, dilution: 1:400), anti-mouse TCRβ-Alexa Fluor 700 (Biolegend, 109224, clone: H57-597, dilution: 1:400), anti-mouse CD3ε-PE/Cyanine7 (Biolegend, 100320, clone: 145-2C11, dilution: 1:400), anti-mouse CD8a-BUV395 (BD Biosciences, 563786, clone: 53-6.7, dilution: 1:400), anti-mouse TCRβ-APC/Cyanine7 (Biolegend, 109220, clone: H57-597, dilution: 1:400), anti-mouse CD3-Alexa Fluor 700 (Biolegend, 100216, clone: 17A2, dilution: 1:400), anti-human/mouse Granzyme B-FITC (Biolegend, 515403, clone: GB11, dilution:1:400), anti-mouse TNFα BV650 (Biolegend, 506333, clone: MP6-XT22, dilution: 1:400), anti-mouse IFNγ-PE/Cyanine7 (Biolegend, 505826, clone: XMG1.2, dilution: 1:400), anti-mouse CD16/CD32 (BioXcell, BE0307, clone: 2.4G2, dilution: 1:400), anti-mouse CD24-BV421 (Biolegend, 101826, clone: M1/69, dilution: 1:400), anti-mouse CD45-BV650 (Biolegend, 103151, clone: 30-F11, dilution: 1:400), anti-mouse CD11c-PE/Cyanine7 (Biolegend, 117318, clone: N418, dilution: 1:400), anti-mouse CD11b-APC-eFluor 780 (eBioscience, 47-0112-82, clone: M1/70, dilution: 1:400), anti-mouse Ly-6C-PerCP/Cyanine5.5 (Biolegend, 128012, clone: HK1.4, dilution: 1:400), anti-mouse I-A/I-E-Alexa Fluor 700 (Biolegend, 107622, clone: M5/114.15.2, dilution: 1:400), anti-mouse CD103-PE (Biolegend, 121406, clone: 2E7, dilution: 1:400), anti-mouse F4/80-Biotin (Biolegend, 123106, clone: BM8, dilution: 1:400), and BUV395 Streptavidin (BD Biosciences, 564176, dilution: 1:1000). Live/dead fixable blue (ThermoFisher, Waltham, Massachusetts, USA) was used to exclude dead cells. For cytokine analysis, cells were incubated for 4 h in RPMI with 10% FBS, phorbol 12-myristate 13-acetate (PMA) (50 ng/ml; Sigma, St. Louis, Missouri, USA), ionomycin (500 ng/ml; Sigma, St. Louis, Missouri, USA) and GolgiStop (BD Biosciences, Franklin Lakes, New Jersey, USA). Cells were stained for surface markers before fixation and permeabilization and then subjected to intracellular cytokine staining according to the manufacturer's protocol (Foxp3 staining buffer set from ThermoFisher, Waltham, Massachusetts, USA). Flow cytometric analysis was performed on an LSR Fortessa (BD Biosciences, Franklin Lakes, New Jersey, USA) and analyzed using FlowJo software (Tree Star, Ashland, Oregon, USA). Immune cell subsets were characterized as follows: leukocytes ($CD45^+$), lymphocytes ($CD45^+CD90.2^+$), $CD4^+$ T cells ($CD45^+CD90.2^+CD4^+CD3^+TCRβ^+$), $CD8^+$ T cells ($CD45^+CD90.2^+CD8a^+CD3^+TCRβ^+$), and NK cells ($CD45^+CD90.2^+NK1.1^+CD3^-$), M1 macrophages ($CD45^+Ly6c^-CD24^-MHCII^+F4/80^+CD11b^+$), M2 macrophages ($CD45^+Ly6c^-CD24^-MHCII^-F4/80^+CD11c^+$), $CD11b^+DC$ ($CD45^+Ly6c^-CD24^+MHCII^+CD11b^+$), $CD103^+DC$ ($CD45^+Ly6c^-CD24^+MHCII^+CD103^+$).

### Flow cytometry analysis of cell surface PD-L1
Cells were dissociated with trypsin and washed twice with PBS with 2% FBS, stained with antibody against mouse PD-L1 (Biolegend, 124311, clone:10 F.9G2, dilution: 1:400) and analyzed on an LSR Fortessa flow cytometer (BD Biosciences, Franklin Lakes, New Jersey, USA).

### qPCR
Total RNA was extracted from cultured cancer cells and tumors using the TRIZOL. RNA was converted into cDNA using 5x ALL-In-One RT Master Mix (Abm, Richmond, British Columbia, Canada) or HiScript III All-in-one RT SuperMix Perfect for qPCR (Vazyme, Nanjing, Jiangsu, China) according to the manufacturer's instructions. Quantitative PCR was performed using SYBR Premix Ex Taq™ (Takara, Kusatsu, Shiga, Japan) or Taq Pro Universal SYBR qPCR Master Mix (Vazyme, Nanjing, Jiangsu, China) in a CFX96 Real-Time System (Bio-Rad, Hercules, California, USA). The primers used were as follows: mouse *Gapdh*, forward 5′-TCACCACCATGGAGAAGGCC-3′, reverse 5′-GCTAAGCAGTTGGTGGTGCAG-3′; mouse *Mc1r*, forward 5′-TGGGCATCATTGCTATAGACCGC-3′, reverse 5′-AACGGCTGTGTG-CTTGTAGTAGG-3′; mouse *Cxcl9*, forward 5′-GTGTGGAGTTCGAGGAACCC-3′, reverse 5′- AATTGGGGCTTGGGGCAAAC-3′; mouse *Cxcl10*, forward 5′-GTCTGAGTGGGACTCA-AGGGAT-3′, reverse 5′-TCTCAACACGTGGGCAGGAT-3′; mouse *Cxcl11*, forward 5′-GCTGCGACAAAGTTGAAGTGATTGTT-3′, reverse

5′-GAGGGCTCACAGTCAGACG-3′; human *ACTIN*, forward 5′-CATGTACGTTGCTATCCAGGC-3′, reverse 5′-CTCCTTAATGTCACGCACGAT-3′; Human *CXCL9*, forward 5′-GTTCTGATTGGAGTGCAAGGAACC-3′, reverse 5′-ATTTTCTCGCAGGAAGGGCTTG-3′; human *CXCL10*, forward 5′- GTCCACGTGTTGAGATCATTGC-3′, reverse 5′-ATCGATTTTGCTCCCCTCTGG-3′; human *CXCL11*, forward 5′-GCTACAGTTGTTCAAGGCTTCCC-3′, reverse 5′-GGAGGCTTTCTCAATATCTGCCAC-3′; Firefly luciferase, forward 5′-GACCGGGACAAAACCATCGC-3′, reverse 5′-TGGCACCACGCTGAGGATAG-3′. All quantitative RT−PCR programs were 95 °C for 3 min, 39 cycles of 95 °C for 10 s and 60 °C for 30 s, followed by a melting curve. Relative mRNA expression was evaluated after normalization for *Gapdh* expression using the standard ΔΔCT method.

### Western blotting
To prepare protein lysates for western blotting, adherent cells were washed with DPBS to remove residual medium and then lysed in SDS lysis buffer (20 mM HEPES, 2 mM $MgCl_2$, 10 mM NaCl, 1% SDS, pH 8.0) freshly supplemented with 0.5 units/μl benzonase, 1× cOmplete, EDTA-free protease inhibitor cocktail (Roche, Basel, Switzerland) and 1× PhosSTOP, Phosphatase inhibitor cocktail (Roche, Basel, Switzerland). Between 5 μg and 30 μg of protein from the samples was separated on a 10% SDS−PAGE gel and transferred to nitrocellulose membranes with a pore size of 0.5 microns. The membranes were blocked in 5% nonfat milk PBST solution (0.1% v/v Tween-20) for 30 min and then incubated with primary antibodies overnight at 4 °C. The following primary antibodies were diluted in 5% nonfat milk PBST: anti-CREB (#9197; Cell Signaling Technology, Danvers, USA; 1:2000), anti-p-CREB (#9198; Cell Signaling Technology, Danvers, USA; 1:2000), anti-Cas9 (#14697; Cell Signaling Technology, Danvers, USA; 1:2000), anti-GNAS (#10150-2-AP; Proteintech, Rosemont, USA; 1:2000), anti-PD-L1 (#ab213480; Abcam, Waltham, USA; 1:2000) and anti-β-actin-HRP (HX18271; Huaxingbio, Beijing, China; 1:10,000). Membranes were washed three times in PBST for 10 min once after blotting with primary antibodies. Then, the membranes were blotted with corresponding secondary antibodies conjugated with HRP enzyme (anti-rabbit IgG, #7074, Cell Signaling Technology, Danvers, USA, 1:10,000 and anti-mouse IgG, #ZB-2305, ZSGB-Bio, Beijing, China, 1:10,000). M5 HiPer ECL Western HRP Substrate (Mei5bio, Beijing, China) was used for the detection of HRP enzymatic activity. Western blot images were taken with a VILBER FUSION FX7 imager.

### ELISA
A total of $0.7 \times 10^6$ cells were seeded per well in 6-well plates and treated with 1 μM α-MSH and/or 10 ng/ml IFNγ for 24 h. Chemokines were measured in supernatants using mouse CXCL9 ELISA Kit (BOSTER) and mouse CXCL10/IP10 ELISA Kit (ABclonal) according to the manufacturer's protocols.

### CRE luciferase assay
A total of $0.1 \times 10^6$ cells were seeded per well in 12-well plates and treated with or without 1 μM α-MSH for 24 h. Cells were dissociated by trypsin and 10% of cells were transferred into 96-well plates. Luciferase activity was measured using Bright-Glo™ Luciferase Assay System (Promega). For normalization, total ATP levels reflecting cell numbers were measured using CellTiter-Glo® Luminescent Cell Viability Assay (Promega) according to the manufacturer's protocol.

### RNA-seq
For RNA sequencing of tumor cells cultured in vitro, B16F10 cells were treated with 1 μM α-MSH (HY-P0252; MedChemExpress, Monmouth Junction, NJ, USA) and 10 ng/ml mouse IFNγ (50709-MNAH; SinoBiological, Beijing, China) for 12 h. For RNA sequencing of tumor cells in vivo, $2 \times 10^6$ ZsGreen-labeled B16F10 cells were subcutaneously injected into wild-type C57BL/6 mice. Ten days after inoculation, tumors were isolated, minced and digested in RPMI-

1640 with 1 mg/ml collagenase I (C2674-1G; Sigma, St. Louis, Missouri, USA) and 200 μg/ml DNase I (DN25-1G; Sigma, St. Louis, Missouri, USA) for 20 min at 37 °C. Single cells were obtained by filtering through a 100-micron cell filter and stained with APC/Cyanine7 labeled anti-CD45 antibody (#103116; Biolegend, San Diego, California, USA; 1:400) in MACS buffer for 20 min at 4 °C. ZsGreen+ CD45- cells were collected by fluorescence-activated cell sorting using a BD FACSAria. Total RNA was extracted from sorted tumor cells using TRIzol, and RNA sequencing was performed by Berry Genomics (Beijing, China). For RNA-seq data analysis, Fastq files were aligned to the mouse reference transcriptome (mm10) using Botwie2 (v2.3.5.1)[78] followed by gene-level quantification with RSEM (v1.3.3)[79]. Differential gene expression and gene ontology analyses were performed with DESeq2 (v1.32.0)[80] and clusterprofiler (v4.0.5)[81], respectively. Volcano plots and heatmaps were generated with R packages ggplot2 (v3.4.0) and pHeatmap (v1.0.12).

## ATAC-seq

A total of $0.5 \times 10^6$ B6F10 cells were seeded per well in 6-well plates and treated with 1 μM αMSH and/or 10 ng/ml IFNγ for 12 h. After treatment, $0.2 \times 10^6$ cells were lysed with the cell lysis buffer (10 mM Tris-HCl pH 7.4, 10 mM NaCl, 3 mM MgCl$_2$, 0.5% NP40). ATAC-seq libraries were prepared using a commercial ATAC kit (Chromatin Profile Kit for Illumina, Novoprotein Inc., Tianjin, China) according to the manufacturer's instructions.

## ChIP-seq

H3K27ac ChIP was performed using the anti-H3K27ac antibody (ab4729, Abcam, Waltham, Massachusetts, USA, 10 μg for 15 million cells). To inhibit histone deacetylase activity, 5 mM sodium butyrate was added to all buffers after cell crosslinking. In brief, a total of $1 \times 10^7$ B16F10 cells were seeded per 15-cm dish and treated with 1 μM αMSH and/or 10 ng/ml IFNγ for 12 h. After treatment, cells were crosslinked with 1% formaldehyde for 10 min at room temperature. Formaldehyde crosslinking was quenched by adding 2 M glycine to achieve a final concentration of 125 mM. After formaldehyde crosslinking, cells were washed twice with ice-cold DPBS and were lysed in lysis buffer 1 (10 mM Tris-HCl pH 7.5, 10 mM NaCl, 0.5% NP40O) freshly supplemented with 1× cOmplete, EDTA-free protease inhibitor cocktail (Roche, Basel, Switzerland) for 10 min on ice. After centrifugation, the cells were resuspended in lysis buffer 2 (20 mM Tris-HCl pH 7.5, 15 mM NaCl, 60 mM KCl, 1 mM CaCl$_2$) freshly supplemented with 1× cOmplete, EDTA-free protease inhibitor cocktail. After centrifugation, pellets were re-resuspended in lysis buffer 2 and digested with micrococcal nuclease (NEB, Ipswich, Massachusetts, USA). A titration of micrococcal nuclease was performed to optimize chromatin fragmentation to mono-nucleosomes. After micrococcal nuclease digestion, an equal volume of lysis buffer 3 (100 mM Tris-HCl pH 8.0, 20 mM EDTA, 200 mM NaCl, 2% Triton X-100, 0.2% sodium deoxycholate, 0.2% SDS) freshly supplemented with 1× cOmplete, EDTA-free protease inhibitor cocktail was added. The resulting samples were sonicated for 10 cycles (high energy, 30 s on, 30 s off) by the Bioruptor Plus sonication device (Diagenode, Denville, NJ, USA). Sonicated chromatin was diluted with an equal volume of lysis buffer 4 (50 mM Tris-HCl pH 8.0, 10 mM EDTA, 100 mM NaCl, 1% Triton X-100, 0.1% sodium deoxycholate) freshly supplemented with 1xcOmplete, EDTA-free protease inhibitor cocktail. After centrifugation, soluble chromatin was incubated with the primary antibody at 4 °C overnight followed by incubation with pre-washed Dynabeads Protein A (10001D, ThermoFisher Scientific, Waltham, Massachusetts, USA) for 2 h. Beads were sequentially washed with the following buffers: twice with lysis buffer 4, twice with high salt buffer (50 mM Tris-HCl pH 8.0, 10 mM EDTA, 500 mM NaCl, 1% Triton X-100, 0.1% sodium deoxycholate), twice with Tris/LiCl buffer (10 mM Tris-HCl pH 8.0, 250 mM LiCl, 0.5% NP40, 0.5% sodium deoxycholate, 1 mM EDTA), and twice with TE buffer (50 mM Tris-HCl

pH 8.0, 10 mM EDTA). Bound chromatin was eluted twice with 50 μL elution buffer (10 mM Tris-HCl pH 8.0, 10 mM EDTA, 150 mM NaCl, 5 mM DTT, 1% SDS) at 65 °C. Eluted chromatin was reverse-crosslinked at 65 °C overnight and digested with RNase A at 37 °C for 1 h, followed by digestion with proteinase K at 65 °C for 2 h. Released DNA was purified with MinElute PCR purification kit (Qiagen, Venlo, Netherlands). ChIP-seq libraries were constructed and sequenced by Beijing Glbizzia Biotechnology Co., Ltd.

## Analysis of ATAC-seq and ChIP-seq data

Adaptors of paired-end reads were trimmed by fastp (v0.23.2). Clean reads were aligned to mouse reference genome (mm10) using Bowtie2 (v2.3.5.1). Peaks were called by MACS2 (v2.2.7.1)[82] with parameter 'macs2 callpeak -g mm --broad --bdg -f BAMPE'. Bigwig files were generated by deepTools (v3.4.3)[83] with the function bamCoverage from BAM files. Read coverage of bigwig files was normalized as Counts Per Million mapped reads (CPM). ChIP-seq and ATAC-seq results were visualized by IGV (v2.12.3)[84] in the bigwig format. Annotation of ChIP-seq and ATAC-seq peaks was achieved by ChIPseeker (v1.28.3)[85] and gene promoter regions were defined as 3.0 kb upstream to 3.0 kb downstream of the annotated TSS. Motif enrichment analysis was performed by HOMER (v4.11)[86] with the function findMotifsGenome.pl.

## TCGA data analysis

TCGA melanoma RNA-seq data were downloaded from UCSC Xena browser[87]. Correlation analyses and heatmap were generated in R. Patients with *GNAS* mutations (R201C/H/L) in TCGA were identified with CBioPortal[88]. Patient survival and differential gene expression were analyzed via CBioPortal[88].

## Derivation of the MAP cell line

Liver cancer was induced by hydrodynamic tail-vein injection as previously described[55]. In brief, a mixture of two plasmids (30 μg each) that collectively encode the piggyBac transposase, Myc, Cas9 and sgRNAs targeting *Apc* and *Tp53* were mixed in 2 ml of normal saline solution and injected via the tail vein into 8-week-old wild-type C57BL/6 mice within 10 s. Approximately 2 months after injection, mouse liver tumors formed, and the resulting tumor blocks were subcutaneously grafted into wild-type mice for growth and immune editing. After three rounds of transplantation, tumors were isolated, minced, and digested in DMEM with 1 mg/ml collagenase I (Sigma, St. Louis, Missouri, USA) and benzonase (Beyotime, Beijing, China) for 30 min at 37 °C. Digested cells were washed three times with PBS and seeded in DMEM supplemented with 10% fetal bovine serum, 2 mM L-glutamine, 0.1% Plasmocin prophylactic, and 1% penicillin–streptomycin. The following sgRNAs were used: *Tp53* sgRNA (5'-CCTCGAGCTCCCTCTGAGCC-3') and *Apc* sgRNA (5'-GGACATGGAGAAGCGTGCAC-3').

## Quantification and statistical analysis

Details of quantification and statistical analysis can be found in the figure legends.

## Reporting summary

Further information on research design is available in the Nature Portfolio Reporting Summary linked to this article.

# Data availability

The RNA-seq, ATAC-seq, and ChIP-seq data generated in this study have been deposited in NCBI GEO with accession GSE214859 and GSE225553. The TCGA publicly available data used in this study are available in the cBioPortal database (https://www.cbioportal.org). The remaining data are available within the Article, Supplementary Information or Source Data file. Source data are provided with this paper.

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

## Acknowledgements

We thank Bing Zhu for helpful discussions, John Snyder for manuscript editing, Qi Li for Cas9 transgenic mice and the A-CREB construct, Hong Zou for help with molecular biology, and Ziye Zou for bioinformatic analysis. We thank BeiGene Co., Ltd., Beijing, China for sharing the anti-PD-1 antibody. This work was supported by Beijing Municipal Commission of Science and Technology (Z201100005320010 and Z22110000 3422015 to T.H.; Z201100005320011 and Z221100003422014 to M.X.), the National Natural Science Foundation of China (No. 82073021 to L.G.), and startup funding from National Institute of Biological Sciences, Beijing (to T.H. and M.X.).

## Author contributions

Ya.C. designed and conducted in vivo screening, tumor challenge, and mechanistic studies; Y.M. analyzed immune infiltration and conducted

mechanistic studies; L.C. analyzed ATAC-seq and ChIP-seq data; L.G. helped with melanoma models, Z.Z. and C.Y. helped with sgRNA design and in vivo screening; T.H., Ya.C. and Yu.C. performed TCGA analysis; T.H. and M.X. obtained funding, designed and supervised the study; T.H., M.X. and Ya.C. wrote the manuscript with input from all authors.

## Competing interests

The authors declare no competing interests.
