## [Peer Review File · Nature Communications]

Activation of melanocortin-1 receptor signaling in melanoma cells impairs T cell infiltration to dampen antitumor immunityEditorial Note: Parts of this Peer Review File have been redacted as indicated to remove third-party material where no permission to publish could be obtained.

REVIEWER COMMENTS

Reviewer #1 (Remarks to the Author): with expertise in CRISPR screen, cancer immunology

Cui et al identify a PKA mediated signalling module that acts as an inhibitory hub for altering the invasion of anti tumourigenic T cells. They identify these axis through an in vitro/ in vivo screen in which they identify the receptor MC1R that regulates this axis. The inhibition of T cell infiltration is regulated through dampening of CXCL9/10/11 secretion from the tumour cells in response to IFN γ .

The experiments are generally well performed and the results validated from many different angles.

I always have an issue of using Cas9 or variants of Cas9 in fully immune competent mice as it is well known that Cas9 is highly immunogenic. While this might not affect the results per se, I would like to see some of the validation work of Figure 1 shown in wt also in Cas9 transgenic animals. While the results might be the same, it would be important to test.

Along the same lines the authors suggest the following, "Because recent studies have revealed the immunogenicity of Cas9 that could influence the antitumor immune response, we monitored dCas9 expression levels by western blotting and found that

MC1R depletion did not affect dCas9 expression (Figure 1E)." I don't understand that argument and I also don't understand the rationale for switching to the dCas9-KRAB system to validate the results from the screen. Could the authors please explain and ideally also do one experiment in which the wt Cas9 plus 2 independent MC1R guide RNAs are used and tested in vivo?

It will also be important to clarify for many experiments (e.g. RNAseq, CXCL9/10 knockout

clones, etc) how many Biological replicates were used and how biological replicate is defined as shown e.g. in Figure 3. The authors need to make sure that at least 2, ideally 3 biological replicates for some cell lines are being examined.

Minor points:

Statistical analysis of Figure 5d is only done between double ko compared to double NTC. Can they also compare the double to the single KO in that experiment?

In Figure 5E it would be important to compare the double to the single ko in the in vivo experiments, e.g. Prkaca sgRNA/ Prkacb sgRNA with Prkaca sgRNA/ NTC or Prkacb sgRNA/NTC.

In Figure 5A expression of GNAS WT and GNAS R201C and in Figure 5F A_CREB should be shown by western analysis.

Reviewer #2 (Remarks to the Author): with expertise in melanoma, Mc1r

In the submitted manuscript, Cui et al. and colleagues performed a CRISPR screening in B16F10 cells. Their results showed that Mc1r impaired T-cell infiltration by inhibiting IFN-gamma-induced CXCL9/10/11 transcription. The authors also provide evidence showing the same mechanism in mouse breast and liver cancer.

Major concerns:

1. The design of the CRISPR screening seems problematic. The authors indicate they used the same method as Manguso, R. et al. Nature 547, 413–418 (2017); however, in the Nature 2017 paper, they used both Tcra^{-/-} and in vitro as controls and treated mice with GVAX or GVAX combined with PD-1 to generate immune-selective pressure on the tumor cells. The authors in this paper only injected the cells into the mice without any treatment; it is doubtful that such a strategy could induce enough immune-selective pressure on the B16 cells. The CRISPR screening in this paper seems to repeat the previous screening with a different library but lower quality.
2. It is not justified how the authors select candidates from the screening result. The

previous screening in the Nature 2017 paper identified multiple members of the same pathway or even the same multi-protein complex. It seems here Mc1r was picked because it has the smallest p-value. Numerous GPCRs can activate cAMP-PKA signaling, and it is not clear whether the screening detected them. It is also surprising that only two known immune invasion regulators were detected.

3. In Figures 1H, 2E, 4F, 4G and S6E, the authors analyzed SKCM TCGA data, with separating groups indicated by MC1R low and high. Such analysis could not draw any sound conclusion because MC1R Red-hair-variant (RHC) significantly impairs the MC1R function, and these variants are frequent in humans. As shown in the previous paper (Robles-Espinoza, C. et al. Nat Commun 7, 12064 (2016)), 8% of individuals had two strong MC1R RHC alleles, and 41% had one in the TCGA SKCM dataset. A sole mRNA expression level could not correctly indicate the function of MC1R. Thus the analysis in Figures 1H, 2E, 4F, and 4G could not support the conclusion.

4. The authors proposed that GNAS-PKA signaling regulates anti-tumor T-cell response. However, the role of GNAS-cAMP-PKA in tumor cell growth is not thoroughly evaluated. In Figure 1G, the authors found that Mc1r knockdown did not affect tumor growth; the mechanism is unknown. It is well known that cAMP-PKA promotes cancer cell proliferation. In this paper (Cancer Cell, 2020 Mar 16;37(3):324-339.e8), a similar in vivo model was used, and the result showed that cAMP-PKA signaling promotes B16F10 growth in vivo but also enhances immune invasion via PD-L1 regulation. Several papers also show that inhibition of MC1R suppresses cell growth (Onco Targets Ther. 2020; 13: 12457–12469., Oncotarget. 2016 May 3; 7(18): 26331–26345.).

5. The authors indicate, "Whereas previous studies have revealed the mechanisms by which aberrant GPCR signaling promote cancer cell proliferation, survival, invasion, and metastasis, their roles in modulating antitumor immunity were unknown." However, the role of GPCR in anti-tumor immunity is well-studied. Examples: ADORA1 inhibition promotes tumor immune evasion via cAMP-PKA-ATF3-PD-L1 (Cancer Cell, 2020 Mar 16;37(3):324-339.e8); Activation of GPER inhibits melanoma and improves response to immune checkpoint blockade via cAMP-Myc-PD-L1 (eLife. 2018; 7: e31770.); cAMP promote anti-tumor immunity in LUAD via cAMP-Myc-PD-L1 (Front Oncol. 2022; 12: 904969.); cAMP induce immune suppression in DLBCL cell lines via regulation of PD-L1 (Leukemia (2021) 35:1990–2001). The involvement of MC1R signaling is also not fully considered in the

manuscript. One of the most well-known downstream targets of MC1R is MITF. Previous research showed that MITF could induce escape from innate immunity in melanoma (J Exp Clin Cancer Res 2021 Mar 31;40(1):117.). Thus the novelty of this research is limited.

6. The role of cAMP-PKA signaling is highly context dependent. Activation of GPCR-cAMP-PKA was shown to enhance anti-tumor immunity in B16F10 cells by reducing PD-L1 (Cancer Cell, 2020 Mar 16;37(3):324-339.e8, eLife. 2018; 7: e31770.). The authors in this research overlooked the involvement of PD-L1 and could not explain why their result is opposite to these findings in melanoma through the same mechanism. cAMP was also shown to promote anti-tumor immunity in LUAD by reducing PD-L1, but suppress anti-tumor immunity in DLBCL cell lines by increasing PD-L1 (Front Oncol. 2022; 12: 904969, Leukemia (2021) 35:1990–2001). The authors could not explain the tissue specific role of cAMP-PKA pathway, cAMP-inhibited CXCL10 production was also reported previously (Brain Res 2015 Jan 12;1594:27-35.).

Minor concerns:

1. Figure 1D, the protein level of Mc1r need to be confirmed for knockdown.
2. Figure 1E and F, it is clearly shown that here the Mc1r function is dependent on a-MSH treatment. However, it is unknown whether Mc1r is fully activated in vivo without any a-MSH treatment.
3. Figure 2A-C, 5C, cell numbers are not enough to determine T cell response. The cells need to be normalized total infiltrating immune cells in the tumors.
4. Figure 2D, although MFI reaches statistical significance, the difference is marginal and it is not sure whether it can cause biological significance.
5. In Figure 3, there is not enough justification to add IFN-gamma in the experiment. Previous research showed that cAMP-PKA suppressed PD-L1 expression and enhanced anti-tumor immunity are not dependent on IFN-gamma.

Reviewer #3 (Remarks to the Author): with expertise in melanoma

In this interesting manuscript, Cui et al. show that the melanocortin receptor-1 (MC1R) promotes B16 murine melanoma growth in immune competent mice via inhibiting expression of CXCL9/10/11. This chemokine reduction is dependent on classical Gs

cAMP/PKA/CREB signaling downstream of MC1R and results in an increased immune cell response. Parallel experiments in immune deficient mice show that the tumor promoting effects of MC1R depletion are observed only in immune competent mice.

Overall this is an intriguing and potentially important report. The data are generally well-presented. However, additional critical controls are needed to validate the findings and interpretation.

1. Most experiments used that dCas9/KRAB repressor protein with gRNA targeting MC1R. The control for this was nontargeting (nonfunctional) gRNA. Experiments should be repeated using a gRNA that is functional and targets a noncritical melanoma gene. The western blot showing that the Cas9 levels in cell lysates are similar is reasonable, but in my view not sufficient to rule out an artifact of the Cas9/KRAB approach. Further, to control for off target effects, it is necessary to rescue the effects of the MC1R depletion with a gRNA resistant MC1R transgene. Similarly, transgene rescue is needed for the Prkaca/Prkabc experiments.

2. It is difficult to determine the significance of the associative human TCGA data without knowing the MC1R genotype. Humans have many common MC1R (hypofunctional) variants, and these are associated with increased risk of melanoma. If the authors' overarching idea is correct, the TCGA data should stratify by MC1R genotype.

3. This paper relies on a single murine cell line model (B16) for all of the melanoma studies. This line lacks the normal oncogene mutations seen in human melanoma and is of questionable physiologic relevance. Other more modern syngeneic melanoma models are now widely available and at least one of them should be used to validate the findings in B16.

4. The well-established MC1R target MITF is associated with improved clinical outcomes in people (PMID 263171710). This is opposite of what would be predicted based on the authors hypothesis. This discrepancy should be discussed.

5. Negative data showing lack of tumor growth differences between MC1R expressing vs.

MC1R depleted B16 cells in cells with concurrent CXCL9/10 depletion is reasonable. However, to fully support the conclusion, these data should be complemented by positive data an experiment showing that expression of a gRNA resistant CXCL9/10 transgene rescues the in vivo tumor phenotype. This would then establish the necessity and sufficiency of CXCL9/10 depletion as the major mediator of the MC1R tumor promoting effects.

6. GNAS(201C) is apparently associated with increased IFN and GZMB positivity in CD8 TILs. This seems opposite to what would be expected given the protumor effect of the constitutively active GNAS mutant. This should be discussed.

7. While I (presumably) understand the rationale for including the small liver and breast studies at the end of the paper, I found them somewhat distracting and unnecessary for this manuscript. They also lack some critical controls, and in my view, the paper would be stronger to use that space to validate the melanoma findings in a different melanoma model.

Reviewer #4 (Remarks to the Author): with expertise in cancer immunology, melanoma

In this manuscript, the authors described a resistant mechanism that melanoma cancer cells express Mc1r to target the IFN-gamma pathway. This leads to the reduction of the expression of CXCL9-11 and therefore the infiltration of T cells. Additional studies indicated that MC1R activates the GNAS-PKA axis to inhibit interferon-gamma (IFN γ) induced CXCL9/10/11 transcription. Together this study reveals MC1R as a potential immunotherapy target that limits antitumor T cell response in melanoma. However, there are many unanswered questions that weaken the claim.

Main:

1. At present, it is still unclear to how important Mc1r expression contributes to immune resistance in melanoma. How broadly is Mc1r expressed in melanoma or other cancers? And what regulates the expression of Mc1r?
2. Screening out Mc1r as a suppressor for T cell infiltration is a big surprise to me: assuming loss of Mc1r leads to the increase of CXCL9/10/11 and the increase of CD8 T cells, how does Mc1r-negative B16F10 cells have the disadvantage over other tumor cells in vivo? This

change of chemokines would impact cancer cells in a whole population, which does not explain its individual disadvantage over its neighboring cancer cells.

3. Only B16F10 was tested for Mc1r function in vivo.

4. How much CXCL9/10 within B16F10 tumor tissues comes from host cells vs cancer cells?

Will loss of Mc1r in cancer cells impact intratumoral CXCL9/10?

Minor:

1. Lack of data about validating the loss of Mc1r protein expression.

2. Why did Mc1r ablation also increase CD4 T or NK cells? And what about other non-T immune cells in the tumors?

Response to reviewers

We are grateful to all four expert reviewers for their insightful comments. We believe we have addressed their comments to the best of our ability resulting in a significantly strengthened manuscript.

In particular, we have performed multiple control experiments to address concerns with respect to the CRISPR screen and Cas9 system related claims. We also have improved the analysis of the TCGA data by stratifying patients according to their *MC1R* genotypes. Moreover, we identified an additional mouse melanoma model (HCmel1274) that depends on MC1R to evade antitumor immunity.

In addition to the point-by-point response to the reviewers' comments, we provide the following table to highlight changes in figures.

Previous submission	This submission	Changes
Figure 1	Figure 1	1H (non-RHC)
Figure 2	Figure 2	2E (non-RHC)
Figure 3	Figure 3	No change
Figure 4	Figure 4	4F (non-RHC), 4G removed
	Figure 5	New data (a new melanoma model HCmel1274)
Figure 5	Figure 6	No change
Figure 6	Figure 7	No change
Figure S1	Figure S1	S1B-H (new data)
Figure S2	Figure S2	S2B (new data)
Figure S3	Figure S3	No change
Figure S4	Figure S4	No change
	Figure S5	New data (PD-L1)
Figure S5	Figure S6	No change
Figure S6	Figure S7	S7A (new data), S7E-F (non-RHC)
Figure S7	Figure S8	S8A (new data)
Figure S8	Figure S9	S9B, D-H (new data)
Figure S9	Figure S10	No change

We are very excited by our new results and we look forward to reviewers' feedback.

In the following point-by-point response, reviewer comments are shown in blue, while our responses are shown in black.

Reviewer #1 (Remarks to the Author): with expertise in CRISPR screen, cancer immunology

Cui et al identify a PKA mediated signalling module that acts as an inhibitory hub for altering the invasion of anti tumourigenic T cells. They identify these axis through an in vitro/ in vivo screen in which they identify the receptor MC1R that regulates this axis. The inhibition of T cell infiltration is regulated through dampening of CXCL9/10/11 secretion from the tumour cells in response to IFNg. The experiments are generally well performed and the results validated from many different angles.

I always have an issue of using Cas9 or variants of Cas9 in fully immune competent mice as it is well known that Cas9 is highly immunogenic. While this might not affect the results per se, I

would like to see some of the validation work of Figure 1 shown in wt also in Cas9 transgenic animals. While the results might be the same, it would be important to test.

We thank the reviewer for pointing out this issue. We performed tumor transplantation experiments in Cas9 transgenic mice as suggested and the results were the same (Figure S1F).

Along the same lines the authors suggest the following, "Because recent studies have revealed the immunogenicity of Cas9 that could influence the antitumor immune response, we monitored dCas9 expression levels by western blotting and found that MC1R depletion did not affect dCas9 expression (Figure 1E)." I don't understand that argument and I also don't understand the rationale for switching to the dCas9-KRAB system to validate the results from the screen. Could the authors please explain and ideally also do one experiment in which the wt Cas9 plus 2 independent MC1R guide RNAs are used and tested in vivo?

We thank the reviewer for pointing this out. The reasons to switch to the dCas9-KRAB system are the following: (1) dCas9-KRAB serves as an orthogonal approach to validate our findings from Cas9-based knockout screening; (2) dCas9-KRAB mediated silencing of *Mc1r* was efficient and uniform; (3) we were unable to obtain an antibody to validate the knockout efficiency of *Mc1r* by Cas9 (the widely used anti-mouse MC1R antibody from Santa Cruz is no longer available), whereas the knockdown of *Mc1r* by dCas9-KRAB could be validated by qPCR. To address the concern raised by the reviewer, we conducted an in vivo experiment using wild-type Cas9 plus two independent *MC1R* guide RNAs (Figure S1D-E). The results were the same as those obtained from the dCas9-KRAB system (Figure 1E and 1G).

It will also be important to clarify for many experiments (e.g. RNAseq, CXCL9/10 knockout clones, etc) how many Biological replicates were used and how biological replicate is defined as shown e.g. in Figure3. The authors need to make sure that at least 2, ideally 3 biological replicates for some cell lines are being examined.

We apologize for the incomplete definition of biological replicates in our previous submission. Biological replicates in Figure 3 were cells grown in different wells derived from the same stable cell line (either control or MC1R-depleted B16F10). *CXCL9/10* knockout clones were two independent clones as judged by distinct indels in *CXCL9/10*. We now included the definition of biological replicates as part of the figure legends.

Minor points:

Statistical analysis of Figure 5d is only done between double ko compared to double NTC. Can they also compare the double to the single KO in that experiment?

Due to the limitation of figure space, we show the full comparisons below. *Cxcl9/10/11* levels were significantly higher in double KO in comparison to single KO.

In Figure 5E it would be important to compare the double to the single ko in the in vivo experiments, e.g. Prkaca sgRNA/ Prkacb sgRNA with Prkaca sgRNA/ NTC or Prkacb sgRNA/NTC.

We performed this in vivo experiment as suggested by the reviewer and found that single KO did not slow tumor growth, consistent with the genetic redundancy between *Prkaca* and *Prkacb* (Figure S9B).

In Figure 5A expression of GNAS WT and GNAS R201C and in Figure 5F A_CREB should be shown by western analysis.

We now show these results as Figure S8A and Figure S9H. In Figure S8A, we first demonstrated the specificity of our anti-GNAS antibody by comparing control vs. GNAS KO cells. However, expression of cDNA encoding GNAS WT or R201C did not result in higher levels of GNAS comparing to control cells. This is consistent with the fact that RIC-8 is a limiting chaperone for nucleotide-free G α -subunit states during biosynthetic protein folding prior to G protein heterotrimer assembly (*Ric-8 proteins are molecular chaperones that direct nascent G protein α subunit membrane association*. Science Signaling. 2011 Nov 22;4(200):ra79.)

Reviewer #2 (Remarks to the Author): with expertise in melanoma, Mc1r

In the submitted manuscript, Cui et al. and colleagues performed a CRISPR screening in B16F10 cells. Their results showed that Mc1r impaired T-cell infiltration by inhibiting IFN-gamma-induced CXCL9/10/11 transcription. The authors also provide evidence showing the same mechanism in mouse breast and liver cancer.

Major concerns:

1. The design of the CRISPR screening seems problematic. The authors indicate they used the same method as Manguso, R. et al. Nature 547, 413–418 (2017); however, in the Nature 2017 paper, they used both Tcr α ^{-/-} and in vitro as controls and treated mice with GVAX or GVAX combined with PD-1 to generate immune-selective pressure on the tumor cells. The authors in this paper only injected the cells into the mice without any treatment; it is doubtful that such a strategy could induce enough immune-selective pressure on the B16 cells. The CRISPR screening in this paper seems to repeat the previous screening with a different library but lower quality.

We agree with the reviewer that the designing of our screening was not as thorough as those presented in Manguso, R. et al. Nature 547, 413–418 (2017). Although we identified known positive regulators of immune evasion (CD47 and PTPN2), we did not attempt to draw conclusions on the pathway mediating immune evasion. Our study was focused on MC1R instead of the entire pathways of immune evasion in the B16F10 melanoma model. We modified the text to reflect the above limitations of our screening.

2. It is not justified how the authors select candidates from the screening result. The previous screening in the Nature 2017 paper identified multiple members of the same pathway or even the same multi-protein complex. It seems here Mc1r was picked because it has the smallest p-value. Numerous GPCRs can activate cAMP-PKA signaling, and it is not clear whether the screening detected them. It is also surprising that only two known immune invasion regulators were detected.

Mc1r was selected because of its smallest *P* value, high ratio of depletion in vivo, and our interest in studying its biology. By RNA-seq, MC1R is the only highly expressed Gas-coupled GPCR that can activate cAMP-PKA signaling in B16F10 (Figure S1C). Therefore, no other GPCR should be detected in our screen. As suggested by the reviewer, we did not apply high

immune-selective pressure on the tumor cells, therefore we only identified the strongest hits among all possible mediators of immune evasion.

3. In Figures 1H, 2E, 4F, 4G and S6E, the authors analyzed SKCM TCGA data, with separating groups indicated by MC1R low and high. Such analysis could not draw any sound conclusion because MC1R Red-hair-variant (RHC) significantly impairs the MC1R function, and these variants are frequent in humans. As shown in the previous paper (Robles-Espinoza, C. et al. Nat Commun 7, 12064 (2016)), 8% of individuals had two strong MC1R RHC alleles, and 41% had one in the TCGA SKCM dataset. A sole mRNA expression level could not correctly indicate the function of MC1R. Thus the analysis in Figures 1H, 2E, 4F, and 4G could not support the conclusion.

We thank the reviewer for pointing out this issue. We have now used the *MC1R* variant annotations from Robles-Espinoza, C. et al. Nat Commun 7, 12064 (2016) to exclude the 8% individuals with two strong MC1R RHC alleles and have redone the bioinformatic analyses. The new set of results (Figures 1H, 2E, 4F, S7E, and S7F) support our overarching idea that high *MC1R* expression in SKCM correlates with poor prognosis, low *CXCL9/10/11* expression, and low T cell response.

4. The authors proposed that GNAS-PKA signaling regulates anti-tumor T-cell response. However, the role of GNAS-cAMP-PKA in tumor cell growth is not thoroughly evaluated. In Figure 1G, the authors found that *Mc1r* knockdown did not affect tumor growth; the mechanism is unknown. It is well known that cAMP-PKA promotes cancer cell proliferation. In this paper (Cancer Cell, 2020 Mar 16;37(3):324-339.e8), a similar in vivo model was used, and the result showed that cAMP-PKA signaling promotes B16F10 growth in vivo but also enhances immune invasion via PD-L1 regulation. Several papers also show that inhibition of MC1R suppresses cell growth (Onco Targets Ther. 2020; 13: 12457–12469., Oncotarget. 2016 May 3; 7(18): 26331–26345.).

The first publication (Cancer Cell, 2020 Mar 16;37(3):324-339.e8) cited by the reviewer focused on the adenosine receptor A1 (ADORA1). ADORA1 is a Gai-coupled GPCR, which not only inhibits adenylyl cyclase to dampen cAMP-PKA signaling, but also activates potassium channels called GIRKs (eLife 2019; 8: e44298). In contrast, MC1R is a Gas-coupled GPCR, which activates adenylyl cyclase to promote cAMP-PKA signaling and does not affect GIRK channels. Therefore, the observation that ADORA1 RNAi reduced cancer cell proliferation does not support the claim that cAMP-PKA promotes cancer cell proliferation. The authors in the publication also showed that ADORA1 RNAi promoted B16F10 immune evasion by upregulating PD-L1 via ATF3. However, whether this regulatory axis was through cAMP-PKA signaling has not been examined (see the dotted line in their graphic abstract). To examine whether MC1R depletion affects PD-L1 levels, we used western blotting and cell surface staining followed by flow cytometry to show that MC1R depletion only modestly reduced cell surface PD-L1 levels by less than 20% (Figure S5). Moreover, If PD-L1 downregulation is indeed the mechanism of MC1R mediated immune evasion, we should not be able to see a synergistic effect between MC1R depletion and anti-PD-1 treatment. Based on the above arguments, this Cancer Cell publication is not inconsistent with our findings and does not comprise the conceptual novelty of our work.

The second publication (Onco Targets Ther. 2020; 13: 12457–12469) showed that a fusion protein α -MSH-PE38KDEL consisting of α -MSH and the bacterial toxin PE38KDEL showed high cytotoxicity on MSH receptor-positive melanoma cells. The third publication (Oncotarget. 2016 May 3; 7(18): 26331–26345.) showed that expression of Agouti signaling protein (ASIP), which is a natural antagonist of MC1R, in B16F10 reduced lung metastasis in mice but did not affect in vitro cell growth. Based on our findings, this result can be interpreted as inhibition of MC1R by ASIP reduces B16F10 fitness in immunocompetent mice. However, the authors did not propose or experimentally test this hypothesis. In conclusion, these two papers neither support the claim that inhibition of MC1R suppresses cell growth nor comprise the novelty of our work.

5. The authors indicate, "Whereas previous studies have revealed the mechanisms by which aberrant GPCR signaling promote cancer cell proliferation, survival, invasion, and metastasis, their roles in modulating antitumor immunity were unknown." However, the role of GPCR in anti-tumor immunity is well-studied. Examples: ADORA1 inhibition promotes tumor immune evasion via cAMP-PKA-ATF3-PD-L1 (Cancer Cell, 2020 Mar 16;37(3):324-339.e8); Activation of GPER inhibits melanoma and improves response to immune checkpoint blockade via cAMP-Myc-PD-L1 (eLife. 2018; 7: e31770.); cAMP promote anti-tumor immunity in LUAD via cAMP-Myc-PD-L1 (Front Oncol. 2022; 12: 904969.); cAMP induce immune suppression in DLBCL cell lines via regulation of PD-L1 (Leukemia (2021) 35:1990–2001). The involvement of MC1R signaling is also not fully considered in the manuscript. One of the most well-known downstream targets of MC1R is MITF. Previous research showed that MITF could induce escape from innate immunity in melanoma (J Exp Clin Cancer Res 2021 Mar 31;40(1):117.). Thus the novelty of this research is limited.

We thank the reviewer for pointing out these previous studies connecting various GPCRs to PD-L1 expression. As stated in the response to major point #4, our results indicate that MC1R mediated immune evasion is independent of PD-L1. Instead, we provided substantial experimental evidence that MC1R mediated immune evasion depends on *CXCL9/10*. Therefore, these prior studies do not comprise the novelty of our study. We agree with the reviewer that our statement that the roles of GPCR in modulating antitumor immunity were unknown was not appropriate. We rephrased this sentence as "Whereas previous studies have revealed the mechanisms by which aberrant GPCR signaling promote cancer cell proliferation, survival, invasion, and metastasis, their roles in modulating antitumor immunity were gaining more attention."

The cited study on MITF (J Exp Clin Cancer Res 2021 Mar 31;40(1):117.) showed that MITF-depleted melanoma cells were susceptible to a T cell-independent immune response. The mechanism described in the study is through MITF-induced expression of ADAM10, a key sheddase that cleaves the MICA/B family of ligands for NK cells. In contrast, MC1R-depleted melanomas are susceptible to T cell-dependent immune response. Thus, loss of MITF does not phenocopy loss of MC1R.

6. The role of cAMP-PKA signaling is highly context dependent. Activation of GPCR-cAMP-PKA was shown to enhance anti-tumor immunity in B16F10 cells by reducing PD-L1 (Cancer Cell, 2020 Mar 16;37(3):324-339.e8, eLife. 2018; 7: e31770.). The authors in this research overlooked the involvement of PD-L1 and could not explain why their result is opposite to these findings in melanoma through the same mechanism. cAMP was also shown to promote anti-tumor immunity in LUAD by reducing PD-L1, but suppress anti-tumor immunity in DLBCL cell lines by increasing PD-L1 (Front Oncol. 2022; 12: 904969, Leukemia (2021) 35:1990–2001). The authors could not explain the tissue specific role of cAMP-PKA pathway, cAMP-inhibited CXCL10 production was also reported previously (Brain Res 2015 Jan 12;1594:27-35.).

We have now included quantitative measurements of PD-L1 expression (Figure S5) which shows that PD-L1 is unlikely to be the downstream effector of MC1R-mediated immune evasion. As elaborated with responses to major points #4 and #5, our results are not inconsistent with these cited studies and the novelty of our work is not compromised by them. As an explanation of tissue specific role of cAMP-PKA pathway, we believe the major determinant is likely the expression and activation of specific GPCRs. In B16F10, MC1R is the only highly expressed G_s-coupled GPCR (Figure S1C), explaining why we could identify MC1R from our CRISPR screening. The report (Brain Res 2015 Jan 12;1594:27-35) showed that cAMP inhibited lipopolysaccharide (LPS)-induced CXCL10 production in primary murine microglia cells, which is a very different context from melanoma immune evasion. Therefore, we don't think this study in any means limits the novelty and significance of our work.

Minor concerns:

1. Figure 1D, the protein level of Mc1r need to be confirmed for knockdown.

After trying several commercial antibodies, we were unable to obtain an antibody to measure the protein levels of MC1R (The widely used MC1R antibody from Santa Cruz was no longer available). That's one of the reasons why we switched to the dCas9-KRAB system (the knockdown of *Mc1r* by dCas9-KRAB could be validated by qPCR). Additionally, we used CREB phosphorylation and CRE-luciferase reporter to measure MC1R's biochemical activity. These experiments provided complementary evidence that we have achieved sufficient depletion of MC1R.

2. Figure 1E and F, it is clearly shown that here the Mc1r function is dependent on α -MSH treatment. However, it is unknown whether Mc1r is fully activated in vivo without any α -MSH treatment.

We thank the reviewer for pointing out this issue. To evaluate whether MC1R is fully activated in vivo without any exogenous α -MSH treatment, we transplanted control or MC1R-depleted B16F10 cells harboring the CRE-luciferase reporter into mice, collected tumors on day 10, and performed qPCR of luciferase mRNA. We found that luciferase mRNAs were expressed at significantly lower levels in MC1R-depleted tumors (Figure S7A). This result indicates MC1R is active in vivo in the absence of exogenous α -MSH treatment.

3. Figure 2A-C, 5C, cell numbers are not enough to determine T cell response. The cells need to be normalized total infiltrating immune cells in the tumors.

In addition to cell numbers, we included the normalized data as Figures S2B.

4. Figure 2D, although MFI reaches statistical significance, the difference is marginal and it is not sure whether it can cause biological significance.

We agree with the reviewer that the MFI differences were modest. We modified our text to down tune the interpretation of the biological significance of these differences.

5. In Figure 3, there is not enough justification to add IFN-gamma in the experiment. Previous research showed that cAMP-PKA suppressed PD-L1 expression and enhanced anti-tumor immunity are not dependent on IFN-gamma.

As elaborated in our responses to major points #4-6, PD-L1 is unlikely to be the downstream mediator of MC1R-mediated immune evasion. Because IFN-gamma is a master cytokine in orchestrating anti-tumor immune response, we examined the impact of MC1R signaling in IFN-gamma response and identified a small subset of IFN-gamma induced genes (including *CXCL9/10/11*) that were repressed by MC1R signaling. We believe these experiments were logically designed.

Reviewer #3 (Remarks to the Author): with expertise in melanoma

In this interesting manuscript, Cui et al. show that the melanocortin receptor-1 (MC1R) promotes B16 murine melanoma growth in immune competent mice via inhibiting expression of *CXCL9/10/11*. This chemokine reduction is dependent on classical Gs cAMP/PKA/CREB signaling downstream of MC1R and results in an increased immune cell response. Parallel experiments in immune deficient mice show that the tumor promoting effects of MC1R depletion are observed only in immune competent mice.

Overall this is an intriguing and potentially important report. The data are generally well-presented. However, additional critical controls are needed to validate the findings and

interpretation.

1. Most experiments used that dCas9/KRAB repressor protein with gRNA targeting MC1R. The control for this was nontargeting (nonfunctional) gRNA. Experiments should be repeated using a gRNA that is functional and targets a noncritical melanoma gene. The western blot showing that the Cas9 levels in cell lysates are similar is reasonable, but in my view not sufficient to rule out an artifact of the Cas9/KRAB approach. Further, to control for off target effects, it is necessary to rescue the effects of the MC1R depletion with a gRNA resistant MC1R transgene. Similarly, transgene rescue is needed for the *Prkaca/Prkabc* experiments.

We thank the reviewer for pointing out this issue. As suggested, we performed the following experiments: (1) WT Cas9 with two control sgRNAs (targeting noncritical melanoma genes *Rosa26* and *H11*) and two independent *Mc1r*-targeting sgRNAs (Figure S1E); (2) Rescue of the effects of the MC1R depletion with an sgRNA resistant MC1R transgene (Figure S1G-H); (3) Rescue of the effects of *Prkaca/Prkabc* knockout with an sgRNA resistant PRKACA transgene (Figure S9D-E).

2. It is difficult to determine the significance of the associative human TCGA data without knowing the MC1R genotype. Humans have many common MC1R (hypofunctional) variants, and these are associated with increased risk of melanoma. If the authors' overarching idea is correct, the TCGA data should stratify by MC1R genotype.

We thank the reviewer for pointing out this issue. We have now used the *MC1R* variant annotations from Robles-Espinoza, C. et al. *Nat Commun* 7, 12064 (2016) to exclude the 8% individuals with two strong MC1R RHC alleles and have redone the bioinformatic analyses. The new set of results support our overarching idea that high *MC1R* expression in SKCM correlates with poor prognosis, low T cell response, and low *CXCL9/10/11* expression.

3. This paper relies on a single murine cell line model (B16) for all of the melanoma studies. This line lacks the normal oncogene mutations seen in human melanoma and is of questionable physiologic relevance. Other more modern syngeneic melanoma models are now widely available and at least one of them should be used to validate the findings in B16.

We thank the reviewer for this suggestion. In 2020, Glenn Merlino and colleagues reported a panel of four syngeneic melanoma models, which represented a variety of molecular and phenotypic subtypes of human melanomas and exhibited diverse range of responses to immune checkpoint blockade (*Nature Medicine* 2020, Pubmed ID: 32284588). We found that one of the four models (M3/HCmel1274, representing the melanocytic subtype of melanomas) expressed high levels of MC1R. As shown in Figure 5, depletion of MC1R from this model slowed tumor growth and enhanced response to anti-PD-1 treatment. Together with our previous findings in B16F10, this result suggests that MC1R could be a target for melanomas with melanocytic features.

4. The well-established MC1R target *MITF* is associated with improved clinical outcomes in people (PMID 263171710). This is opposite of what would be predicted based on the authors hypothesis. This discrepancy should be discussed.

In the literature, the association between *MITF* with melanoma clinical outcomes has not reached consensus. In the paper cited by the reviewer (*Melanoma Res.* 2015 Dec;25(6):496-502), the authors showed that *MITF* is associated with improved clinical outcomes in people. However, another highly cited study (*Nature* 436, 117–122 (2005). <https://doi.org/10.1038/nature03664>) showed that *MITF* amplification was more prevalent in malignant melanoma and correlated with decreased overall patient survival.

5. Negative data showing lack of tumor growth differences between MC1R expressing vs. MC1R depleted B16 cells in cells with concurrent *CXCL9/10* depletion is reasonable. However, to fully support the conclusion, these data should be complemented by positive data an experiment

showing that expression of a gRNA resistant CXCL9/10 transgene rescues the in vivo tumor phenotype. This would then establish the necessity and sufficiency of CXCL9/10 depletion as the major mediator of the MC1R tumor promoting effects.

We appreciate the reviewer's suggestion for this experiment. However, CXCL9/10 are inducible genes regulated by multiple signaling pathways, such as IFN γ and cAMP-PKA (as shown by us in this study). It is thus technically challenging to design a transgene construct that can faithfully mimic the spatial and temporal regulation of CXCL9/10 expression.

6. GNAS(201C) is apparently associated with increased IFN and GZMB positivity in CD8 TILs. This seems opposite to what would be expected given the protumor effect of the constitutively active GNAS mutant. This should be discussed.

We thank the reviewer for pointing out the mistake we made in the previous manuscript, which has been corrected in the current manuscript. As shown in Figure S8E, GNAS (201C) was associated with decreased IFN γ and GZMB positivity in CD8 TILs.

7. While I (presumably) understand the rationale for including the small liver and breast studies at the end of the paper, I found them somewhat distracting and unnecessary for this manuscript. They also lack some critical controls, and in my view, the paper would be stronger to use that space to validate the melanoma findings in a different melanoma model.

We thank the reviewer for this suggestion. Because GNAS mutations occur in a variety of tumor types, we feel it is important to examine the effect of GNAS mutations in driving immune evasion in other cancer types. As suggested by the reviewer, we added one entire figure (Figure 5) to validate the B16F10 findings in a different melanoma model HcMel1274.

Reviewer #4 (Remarks to the Author): with expertise in cancer immunology, melanoma

In this manuscript, the authors described a resistant mechanism that melanoma cancer cells express Mc1r to target the IFN- γ pathway. This leads to the reduction of the expression of CXCL9-11 and therefore the infiltration of T cells. Additional studies indicated that MC1R activates the GNAS-PKA axis to inhibit interferon- γ (IFN γ) induced CXCL9/10/11 transcription. Together this study reveals MC1R as a potential immunotherapy target that limits antitumor T cell response in melanoma. However, there are many unanswered questions that weaken the claim.

Main:

1. At present, it is still unclear to how important Mc1r expression contributes to immune resistance in melanoma. How broadly is Mc1r expressed in melanoma or other cancers? And what regulates the expression of Mc1r?

In the new Figure 5A, by analyzing TCGA and GTEx datasets, we show that high MC1R expression was restricted to a subset of melanomas. We currently do not know what regulates the expression of MC1R, but we think this is an interesting question for future investigation.

2. Screening out Mc1r as a suppressor for T cell infiltration is a big surprise to me: assuming loss of Mc1r leads to the increase of CXCL9/10/11 and the increase of CD8 T cells, how does Mc1r-negative B16F10 cells have the disadvantage over other tumor cells in vivo? This change of chemokines would impact cancer cells in a whole population, which does not explain its individual disadvantage over its neighboring cancer cells.

We thank the reviewer for raising this question. Chemokines such as CXCL9/10/11 encodes spatial information to guide the migration of immune cells. Increased CXCL9/10/11 expression in MC1R-null cells are more likely to attract T cells to their vicinity, resulting in a survival disadvantage in vivo. Although neighboring cancer cells may also be attached by T cells as

bystanders, they should not be associated with the loss of specific genes, therefore invisible in the CRISPR screening.

3. Only B16F10 was tested for *Mc1r* function in vivo.

We now devoted one entire figure (Figure 5) to validate the function of MC1R in a different melanoma model HCmel1274.

4. How much CXCL9/10 within B16F10 tumor tissues comes from host cells vs cancer cells? Will loss of *Mc1r* in cancer cells impact intratumoral CXCL9/10?

We thank the reviewer for this question. By qPCR of tumor-derived RNAs, we now demonstrate that loss of MC1R in cancer cells indeed impacted intratumoral CXCL9/10 levels (Figure S7A).

Minor:

1. Lack of data about validating the loss of *Mc1r* protein expression.

After trying several commercial antibodies, we were unable to obtain an antibody to measure the protein levels of MC1R (The widely used MC1R antibody from Santa Cruz was no longer available). That's one of the reasons why we switched to the dCas9-KRAB system (the knockdown of *Mc1r* by dCas9-KRAB could be validated by qPCR). Additionally, we used CREB phosphorylation and CRE-luciferase reporter to measure MC1R's biochemical activity. These experiments provided complementary evidence that we have achieved sufficient depletion of MC1R.

2. Why did *Mc1r* ablation also increase CD4 T or NK cells? And what about other non-T immune cells in the tumors?

MC1R ablation causes an increase of CXCL9/10/11 production. The receptor of these chemokines, CXCR3, is expressed in CD8+ T cells, CD4+ T cells, and NK cells. As shown in Figure S3B, the number of myeloid cells were not affected by MC1R depletion.

REVIEWERS' COMMENTS

Reviewer #1 (Remarks to the Author):

I am satisfied with the revision experiments and have no further requests.

Reviewer #2 (Remarks to the Author):

The authors agreed that the design of their screening was not as thorough as those presented in Manguso, R. et al. *Nature* 547, 413–418 (2017). The authors still did not satisfactorily explain the difference between these two screenings: why the authors can detect the cAMP-PKA signaling pathway as the most substantial factor while Manguso, R. et al. did not with more thorough conditions. The authors indicated they did not attempt to draw conclusions on the pathways; their study was focused on MC1R instead of the entire pathways. However, since the authors provide a lot of results to demonstrate that the effect is mediated through cAMP-PKA, it is pretty surprising not a single other cAMP-PKA mediator was identified through the screening. If the authors say Mc1r is the only highly expressed G α s-coupled GPCR (Figure S1C) in B16F10, it is not meaningful to screen such a protein from other non-expressed candidates. The authors also indicate other GPCRs, such as G α i-coupled GPCR, can activate cAMP-PKA signaling, but these are also not detected in the screening. The authors excluded the 8% of individuals with two strong MC1R RHC alleles in the revised manuscript for the TCGA data analysis, but they should compare the difference among RR, RO, and OO groups as shown in the Robles-Espinoza, C. et al. *Nat Commun* 7, 12064 (2016) paper. For the *Cancer Cell*, 2020 Mar 16;37(3):324-339.e8 article, the authors indicate “this regulatory axis was through cAMP-PKA signaling has not been examined”, but the *Cancer Cell* paper provided comprehensive results to demonstrate that the regulation is through cAMP/PKA/CREB/ATF3/PD-L1 axis (Figure S7). The authors used “Activation of GPCR-GNAS-PKA signaling in cancer cells impairs T cell infiltration to dampen antitumor immunity” as the title; as mentioned in previous comments, the fact in the title has been well studied and widely reported, at least six high-impact papers showed that cAMP-PKA signaling affects antitumor immunity via PD-L1 regulation in melanoma and other cancers. Even the authors presented data to show that MC1R depletion only modestly reduced cell surface PD-L1 levels by less than 20%; the authors should also conduct experiments to show

why the same mechanism (cAMP-PKA signaling) did not lead to the same result as many previous papers. Although the minor concerns were adequately addressed, the major concerns regarding novelty and inconsistencies with previous literature were not sufficiently resolved.

Reviewer #3 (Remarks to the Author):

This revised manuscript includes quite a lot of new data including multiple transgene rescue experiments and in vivo studies using an additional syngeneic melanoma model. These significantly strengthen the paper.

However, as highlighted in the initial review, the author's claim that CXCL9/10/11 suppression is the mechanism by which MC1R signaling promotes melanoma growth in vivo is not definitely established and the authors response to my original comment on that is not particularly satisfying. Authors have clearly demonstrated the ability to knock out and overexpress genes of interest in these melanoma models. It is not clear why transgene expression/restoration of CXCL9/10/11 was not at least attempted. This seems a straightforward experiment. If CXCL9/10 expression reversed the anti-tumor effect of MC1R depletion, then the claim that CXCL9/10 suppression is the main mechanism for their phenotype would be much stronger. As currently presented in Figure 4, melanomas with CXCL9/10 knock out grow slower than melanomas derived from the parental cells. This is opposite to what would be predicted by the authors claims, as the CXCL9/10 knockout tumors should be relatively free from immune cells and thereby free to proliferate rapidly. If this rescue/overexpression experiment was attempted and results were unexpected, the data could still be included in the supplemental figures, and experimental caveats surrounding timing and dose of constitutively active transgenes could be considered in the discussion. In my view, the main phenotype is so dramatic that the manuscript would still be a significant advance, even if there was some remaining uncertainty regarding the mechanism.

Reviewer #4 (Remarks to the Author):

Though the authors had replied to all my questions, I am still not comfort with their response to the advantage of Mc1r-null cells over other targeted cells in vivo in their screening model: once T cells migrate to tumor sites because of increased CXCL9/10, other tumor cells will be recognized and killed by infiltrated T cells at a similar extent. They are not neglected.

Reviewer #1 (Remarks to the Author)

I am satisfied with the revision experiments and have no further requests.

Reviewer #2 (Remarks to the Author)

The authors agreed that the design of their screening was not as thorough as those presented in Manguso, R. et al. Nature 547, 413–418 (2017). The authors still did not satisfactorily explain the difference between these two screenings: why the authors can detect the cAMP-PKA signaling pathway as the most substantial factor while Manguso, R. et al. did not with more thorough conditions. The authors indicated they did not attempt to draw conclusions on the pathways; their study was focused on MC1R instead of the entire pathways. However, since the authors provide a lot of results to demonstrate that the effect is mediated through cAMP-PKA, it is pretty surprising not a single other cAMP-PKA mediator was identified through the screening.

We thank the reviewer for raising this concern. First, downstream components of the cAMP-PKA signaling pathway were not identified in either our screening or those presented in Manguso, R, et al. Nature 2017. The explanation as shown in our paper (Figures 6D-G, S9) is due to genetic redundancy between components of the pathway. There are two PKA paralogs and multiple CREB paralogs in mice. Thus single-gene knockout of these downstream components does not impair cAMP-PKA signaling. Second, *Mc1r* was not identified in Manguso, R, et al. simply because their sgRNA library (9,872 sgRNAs targeting 2,368 genes) did not contain *Mc1r* (see attached excel spreadsheet). While our manuscript was under peer review, the Manguso lab published another in vivo screening paper (Dubrot et al. In vivo CRISPR screens reveal the landscape of immune evasion pathways across cancer. Nature Immunology 2022). In this new study, they used genome-scale in vivo CRISPR screens across eight transplantable mouse tumor models (including B16F10) and provided data in an interactive public data portal (www.tumorimmunity.org). We thus queried this new dataset. Consistent with our findings, *Mc1r* was depleted in wildtype mice relative to in vitro cultured cells (see screenshots from <https://tumorimmunity.org> below). Thus, our finding is now supported by independent genome-scale in vivo CRISPR screening.

[Screenshot redacted]

If the authors say Mc1r is the only highly expressed Gas-coupled GPCR (Figure S1C) in B16F10, it is not meaningful to screen such a protein from other non-expressed candidates.

We thank the reviewer for this comment, which we respectfully disagree. Specific expression of GPCRs makes a substantial contribution to GPCR biology, thus screening out MC1R in the context of immune evasion of melanocytic melanoma is meaningful.

The authors also indicate other GPCRs, such as Gai-coupled GPCR, can activate cAMP-PKA signaling, but these are also not detected in the screening.

We thank the reviewer for this comment. Gai-coupled GPCRs inactivate cAMP-PKA signaling instead of activating cAMP-PKA signaling, thus are not expected to be detected in our screening.

The authors excluded the 8% of individuals with two strong MC1R RHC alleles in the revised manuscript for the TCGA data analysis, but they should compare the difference among RR, R0, and 00 groups as shown in the Robles-Espinoza, C. et al. Nat Commun 7, 12064 (2016) paper.

We thank the reviewer for this comment. We excluded 8% of individuals with two strong MC1R RHC alleles (RR) because they lack functional MC1R and thus their melanomas should be independent of MC1R. Moreover, the relatively small number (n=23) of patients with RR alleles renders comparisons under powered to make scientifically sound conclusions. In any case, we performed survival analysis among

RR, R0, and 00 groups. Higher MC1R expression displays a trend of poor prognosis in R0 and 00 groups but not in 00 groups (with the caveat of under-powered statistical analysis).

For the Cancer Cell, 2020 Mar 16;37(3):324-339.e8 article, the authors indicate “this regulatory axis was through cAMP-PKA signaling has not been examined”, but the Cancer Cell paper provided comprehensive results to demonstrate that the regulation is through cAMP/PKA/CREB/ATF3/PD-L1 axis (Figure S7).

In Figure S7 of the Cancer Cell paper, the authors showed that treatment of A375 and SK-MEL-28 cells with DX (a ADORA1 antagonist) raises cAMP level and PKA activity, increases pCREB and ATF3 levels. Because these experiments did not use genetic or pharmacological tools to block cAMP production or PKA activity, we stated that “this regulatory axis (ADORA1-ATF3-PD-L1) was through cAMP-PKA signaling has not been examined” in our response letter. We are happy to modify this statement as “this regulatory axis was through cAMP-PKA signaling has not been examined by genetic or pharmacologic perturbations of cAMP-PKA signaling”.

The authors used “Activation of GPCR-GNAS-PKA signaling in cancer cells impairs T cell infiltration to dampen antitumor immunity” as the title; as mentioned in previous comments, the fact in the title has been well studied and widely reported, at least six high-impact papers showed that cAMP-PKA signaling affects antitumor immunity via PD-L1 regulation in melanoma and other cancers. Even the authors presented data to show that MC1R depletion only modestly reduced cell surface PD-L1 levels by less than 20%; the authors should also conduct experiments to show why the same mechanism (cAMP-PKA signaling) did not lead to the same result as many previous papers. Although the minor concerns were adequately addressed, the major concerns regarding novelty and inconsistencies with previous literature were not sufficiently resolved.

We thank the reviewer for this comment. We observed modestly reduced cell surface PD-L1 levels upon MC1R depletion, which is not inconsistent with previously published papers showing cAMP-PKA signaling affects PD-L1 expression. Moreover, it is widely appreciated that tumor immune evasion can be driven by diverse mechanisms. The repression of CXCL9/10 by MC1R/cAMP/PKA signaling is more prominent than PD-L1 repression in B16F10, and thus drives immune evasion downstream of MC1R/cAMP/PKA signaling in this model.

Reviewer #3 (Remarks to the Author)

This revised manuscript includes quite a lot of new data including multiple transgene rescue experiments and in vivo studies using an additional syngeneic melanoma model. These significantly strengthen the paper.

However, as highlighted in the initial review, the author's claim that CXCL9/10/11 suppression is the mechanism by which MC1R signaling promotes melanoma growth in vivo is not definitely established and the authors' response to my original comment on that is not particularly satisfying. Authors have clearly demonstrated the ability to knock out and overexpress genes of interest in these melanoma models. It is not clear why transgene expression/restoration of CXCL9/10/11 was not at least attempted. This seems a straightforward experiment. If CXCL9/10 expression reversed the anti-tumor effect of MC1R depletion, then the claim that CXCL9/10 suppression is the main mechanism for their phenotype would be much stronger. As currently presented in Figure 4, melanomas with CXCL9/10 knock out grow slower than melanomas derived from the parental cells. This is opposite to what would be predicted by the authors' claims, as the CXCL9/10 knockout tumors should be relatively free from immune cells and thereby free to proliferate rapidly. If this rescue/overexpression experiment was attempted and results were unexpected, the data could still be included in the supplemental figures, and experimental caveats surrounding timing and dose of constitutively active transgenes could be considered in the discussion. In my view, the main phenotype is so dramatic that the manuscript would still be a significant advance, even if there was some remaining uncertainty regarding the mechanism.

We thank the reviewer for this comment. The reviewer commented: "If this rescue/overexpression experiment was attempted and results were unexpected, the data could still be included in the supplemental figures, and experimental caveats surrounding timing and dose of constitutively active transgenes could be considered in the discussion". For this exact consideration, we believe this rescue/overexpression experiment is of limited value. However, we are willing to do this experiment if the reviewer and the Editor insist on this point.

Reviewer #4 (Remarks to the Author)

Though the authors had replied to all my questions, I am still not comfortable with their response to the advantage of *Mc1r*-null cells over other targeted cells in vivo in their screening model: once T cells migrate to tumor sites because of increased CXCL9/10, other tumor cells will be recognized and killed by infiltrated T cells to a similar extent. They are not neglected.

We thank the reviewer for the request to further clarify this issue. Indeed *Mc1r*-depleted cells attract T cells to their neighborhood to kill both *Mc1r*-depleted cells and bystander cells. However, bystander cells are not associated with any specific gene deletions. Thus, *Mc1r*-depleted cells showed a disadvantage over cells with other gene deletions (see the cartoon below).

- *Mc1r* KO
- Gene A KO
- Gene B KO
- Bystander
- X Killing by T cells